# An IS-mediated, RecA-dependent, bet-hedging strategy in *Burkholderia thailandensis*

Lillian C Lowrey, Leslie A Kent, Bridgett M Rios, Angelica B Ocasio, Peggy A Cotter*

Department of Microbiology and Immunology, University of North Carolina at Chapel Hill, Chapel Hill, United States

**Abstract** Adaptation to fluctuating environmental conditions is difficult to achieve. Phase variation mechanisms can overcome this difficulty by altering genomic architecture in a subset of individuals, creating a phenotypically heterogeneous population with subpopulations optimized to persist when conditions change, or are encountered, suddenly. We have identified a phase variation system in *Burkholderia thailandensis* that generates a genotypically and phenotypically heterogeneous population. Genetic analyses revealed that RecA-mediated homologous recombination between a pair of insertion sequence (IS) *2*-like elements duplicates a 208.6 kb region of DNA that contains 157 coding sequences. RecA-mediated homologous recombination also resolves merodiploids, and hence copy number of the region is varied and dynamic within populations. We showed that the presence of two or more copies of the region is advantageous for growth in a biofilm, and a single copy is advantageous during planktonic growth. While IS elements are well known to contribute to evolution through gene inactivation, polar effects on downstream genes, and altering genomic architecture, we believe that this system represents a rare example of IS element-mediated evolution in which the IS elements provide homologous sequences for amplification of a chromosomal region that provides a selective advantage under specific growth conditions, thereby expanding the lifestyle repertoire of the species.

*For correspondence:
pcotter@med.unc.edu

**Competing interest:** The authors declare that no competing interests exist.

## Editor's evaluation

This paper reports a bet hedging strategy in bacteria based on chromosomal duplications and rearrangements that confer advantages in certain growth conditions. The work is of fundamental importance for understanding the role of genetic and biological variation in bacteria. The experimental work is exceptionally strong and convincing. The paper will be of interest to a broad audience including bacteriologists, geneticists and evolutionary biologists.

## Introduction

Microbes often exist in complex communities wherein productivity and resilience rely on population diversity. This diversity commonly appears as mixed-species populations. However, diversity can also exist within otherwise clonal populations through genotypic and phenotypic heterogeneity. Heterogeneity can increase the fitness of a population by contributing to the division of labor, or it can serve as a bet-hedging strategy to increase the odds of population survival when alteration of environmental conditions, such as nutrient availability, host response, or antimicrobial concentrations, occur quickly (*Magdanova and Golyasnaya, 2013*).

**eLife digest** Bacterial populations are often diverse, even when originating from a single cell. This diversity helps microbes survive in fluctuating environmental conditions by increasing the odds of population survival. For example, if environmental conditions change such that only a subpopulation with unique abilities survives, the entire population will be saved.

Genomes are naturally dynamic. For example, mobile sections of DNA, called transposable elements, can change their position within a genome. If a transposable element jumps into a gene, it can harm the cell. But if it moves into a different site, it may provide an organism with new features that can help it survive.

Most organisms contain multiple copies of transposable elements in their DNA. For example, a subtype of the soil bacterium *Burkholderia thailandensis*, strain E264, has two identical transposable elements that book-end a region of DNA that contains 157 genes. Lowrey et al. studied this bacterial strain in different environmental conditions to find out more. The experiments revealed that in growing populations of E264, some bacteria had one copy of the region, while others had two or three. In a rich environment, most bacteria had just one copy of the region. However, when grown in challenging conditions, most bacteria contained two or three copies of the region.

Moreover, bacteria required at least two copies to form dense communities known as biofilms, which are advantageous for bacterial survival in challenging conditions. Bacteria with only one copy, however, were better adapted to a free-swimming lifestyle.

Lowrey et al. further showed that the DNA repair system was required for duplicating the region. Usually, this system finds and recombines identical DNA sequences to repair broken DNA. However, if two identical DNA sequences (a pair of transposable elements) are present, the repair system can recombine them during DNA replication, resulting in the duplication of the DNA between the identical sequences. The same system also reduces the copy number of the region from three or two to just one.

Since the repair system is constantly working and DNA recombination is always occurring at a low level, *B. thailandensis* E264 maintains a genetically diverse population with bacteria containing different copy numbers of the region. This diversity ensures that the strain survives in fluctuating environmental conditions.

Transposable elements are hotspots of evolution. They are known to interrupt genes and shrink genomes. Lowrey et al. showed that transposable elements also influence evolution by providing DNA sequences that the DNA repair system can use to duplicate DNA. This process of duplicating genes is more frequent than random genetic mutations, expediting adaptation.

A common facilitator of phenotypic heterogeneity is phase variation, in which reversible genetic alterations occur that function as 'ON/OFF' switches that control specific phenotypes (*van der Woude, 2011*; *van der Woude and Bäumler, 2004*). These genetic alterations typically occur at frequencies that are orders of magnitude greater than spontaneous mutations. One common generator of phenotypic heterogeneity is recombinase-mediated, site-specific recombination, such as invertible switches and transposable elements, which can influence the expression of coding sequences by altering the orientation of promoters or introducing new regulatory sequences, respectively (*Trzilova and Tamayo, 2021*; *Schneider and Lenski, 2004*; *Darmon and Leach, 2014*).

Phenotypic heterogeneity within a population can also be achieved by recombination between homologous sequences at different genomic loci, a phenomenon referred to as unequal or non-allelic recombination. Unlike site-specific recombination, which relies on a specific recombinase catalyzing strand exchange between specific target sequences, any homologous sequences greater than 50 bp can act as substrates for unequal homologous recombination. These homologous sequence substrates are often ribosomal RNA operons or transposable elements, which are commonly found in multiple copies per genome (*Andersson and Hughes, 2009*; *Raeside et al., 2014*). The DNA repair and maintenance recombinase, RecA, identifies homologous sequences and catalyzes strand exchange which, when unequal homologous sequences are recombined, leads to modifications in genomic architecture through sequence inversions, deletions, or duplications (*Darmon and Leach, 2014*; *Bourque et al., 2018*). These adjustments to genome architecture can similarly influence the orientation of

promoters through sequence inversion, but also change copy number of genes by deleting and duplicating intervening sequences (*Vandecraen et al., 2017*). Changes in gene copy number can impact protein production, and thus cell phenotypes – a phenomenon referred to as gene dosage effects (*Andersson and Hughes, 2009*; *Sandegren and Andersson, 2009*).

Phenotypic heterogeneity has been observed in a wide range of *Burkholderia* species. The bioterrorism agent *B. pseudomallei* displays at least seven distinct morphologies when propagated on Ashdown's agar, and virulence-associated phenotypes, such as intracellular survival, colonization ability, and antigen presentation, differ between the various morphotypes (*Chantratita et al., 2007*; *Tandhavanant et al., 2010*; *Austin et al., 2015*; *Wikraiphat et al., 2015*). The cystic fibrosis-associated opportunistic pathogen *B. cenocepacia* similarly exhibits phenotypic and genotypic heterogeneity during growth in the lab as well as during growth in the cystic fibrosis lung (*Häussler et al., 2003*; *Lee et al., 2017*; *Springman et al., 2009*).

*Burkholderia thailandensis* strain E264 (*Bt*E264) displays distinct phenotypes, including aggregation, biofilm formation, Congo Red binding, and the production of a gold-brown 'pigment', that were attributed initially to the activity of the *bcpAIOB*-encoded contact-dependent inhibition system and hence we named them contact-dependent signaling phenotypes (*Anderson et al., 2012*; *Garcia et al., 2016*). Subsequent analyses, however, showed that these phenotypes occur when a 208.6 kb region on chromosome I that contains 157 predicted coding sequences plus one of the flanking IS*2*-like elements (shown in *Figure 1*) is present in two or more copies (*Ocasio and Cotter, 2019*), so we will now refer to those phenotypes as duplication-dependent phenotypes. Our previous data suggested that amplification of the 208.6 kb region required BcpAIOB activity and the IS*2*-like element at the 5′ end (ISβ), and that the region amplified as extrachromosomal, circular DNA molecules (*Ocasio and Cotter, 2019*). Our investigations into the mechanism underlying the amplification, however, caused us to completely re-evaluate our previous conclusions. Here, we present data indicating that *Bt*E264 possesses a homologous recombination-mediated phase variation mechanism that facilitates rapid adaptation to disparate growth conditions by altering the copy number of the 208.6 kb region on chromosome I in a manner that is independent of both ISβ and BcpAIOB activity.

## Results

### *Burkholderia thailandensis* E264 from ATCC is a genotypically heterogeneous population

Our investigations led us to suspect that not all of the bacteria in our stock of wild-type *Bt*E264 from ATCC contained multiple copies of the 208.6 kb DNA region. To assess the potential heterogeneity of our stock, we conducted PCR analyses on individual colonies after plating an aliquot of the stock on LSLB agar. To detect the presence of multiple copies of the 208.6 kb region, we use outward-facing primers that anneal near the ends of the 208.6 kb region, but not within the IS elements (*Figure 1A*, Junc1 and Junc2). PCR with these primers yields a 2.2 kb DNA product containing one IS element flanked by the 5′ and 3′ ends of the region if two or more copies of the region are present (*Ocasio and Cotter, 2019*). While a product of the expected 1 kb size was amplified from all colonies when control primers (*Figure 1A*, Ctrl 1 and Ctrl 2) were used (*Figure 1B*), the 2.2-kb 'junction PCR' product was obtained from only approximately 50% of the colonies (*Figure 1B*). These results indicate that our *Bt*E264 stock from ATCC is a genotypically heterogeneous population with approximately half of the bacteria containing a single copy of the 208.6 kb region and half containing two or more copies of the 208.6 kb region.

To determine if this genotypic heterogeneity is unique to our stock of *Bt*E264, we conducted PCR analyses on colonies of *Bt*E264 that we obtained from three separate laboratories. Junction PCR products of the expected 2.2 kb size were obtained for 0/12, 6/12, and 6/12 colonies from *Bt*E264 obtained from the Mougous, Chandler, and Manoil laboratories, respectively, while all colonies yielded 1 kb PCR products when the control primers were used (*Figure 1C–E*). We suspect that the difference in stocks between laboratories can be explained by how the laboratories prepared their wild-type frozen stocks. Each laboratory had obtained its sample of *Bt*E264 from ATCC, suggesting that the ATCC stock of *Bt*E264 is a genotypically heterogeneous population composed of bacteria with either a single ('junction PCR negative') or multiple ('junction PCR positive') copies of the 208.6 kb region.

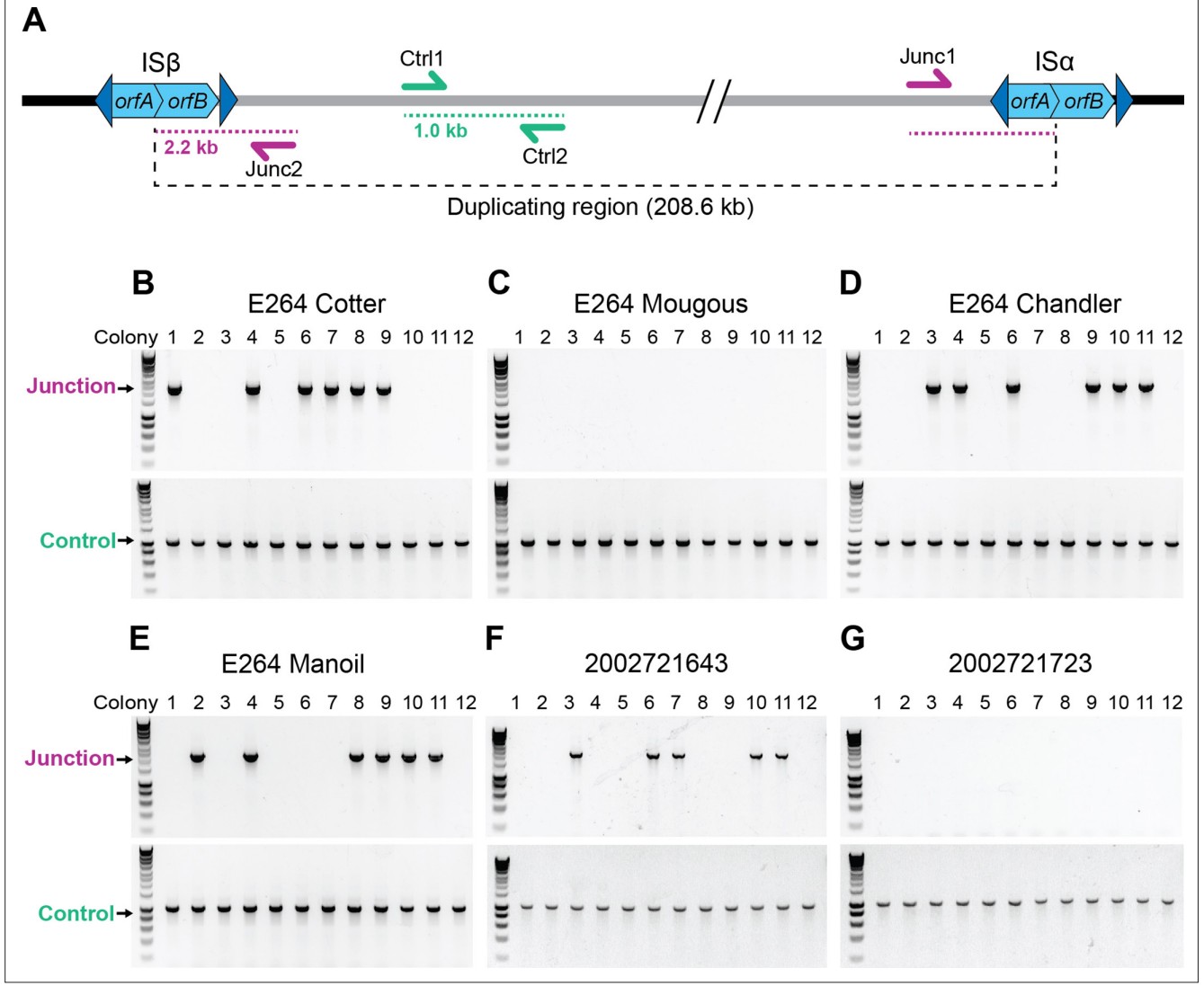

**Figure 1.** *Burkholderia thailandensis* E264 from ATCC is a genotypically heterogeneous population. (**A**) Graphical representation of the 208.6 kb duplicating region bounded by homologous IS*2*-like elements referred to as ISβ (LO74_RS09425) and ISα (LO74_RS28540). The Ctrl1 and Ctrl2 primers amplify a 1 kb DNA product. Junc1 and Junc2 primers amplify a 2.2 kb product if the region has duplicated. (**B–E**) Twelve colonies from stocks of E264 from four different laboratories (Cotter, Mougous, Chandler, and Manoil) were used as templates for PCR using either Junction (Junc1 and Junc2) or Control (Ctrl1 and Ctrl2) primers. (**F, G**) Twelve colonies from *B. thailandensis* strain 2002721643 and 2002721723 were used as templates for PCR using either Junction (Junc1 and Junc2) or Control (Ctrl1 and Ctrl2) primers.

The online version of this article includes the following source data for figure 1:

**Source data 1.** Uncropped DNA gel displaying junction PCR that was used to make *Figure 1B*.

**Source data 2.** Uncropped DNA gel displaying control PCR that was used to make *Figure 1B, C*.

**Source data 3.** Uncropped DNA gel displaying junction PCR that was used to make *Figure 1C*.

**Source data 4.** Uncropped DNA gel displaying junction PCR that was used to make *Figure 1D*.

**Source data 5.** Uncropped DNA gel displaying control PCR that was used to make *Figure 1D, E*.

**Source data 6.** Uncropped DNA gel displaying junction PCR that was used to make *Figure 1E*.

**Source data 7.** Uncropped DNA gel displaying junction PCR that was used to make *Figure 1F, G*.

**Source data 8.** Uncropped DNA gel displaying control PCR that was used to make *Figure 1F, G*.

A small number of other *B. thailandensis* strains for which genome sequence is available also possess an IS-bounded region similar to that in *Bt*E264. Two of these strains, 2002721643 and 2002721723, have genomic synteny of 99.0% and 98.9% and region synteny of 96.2% and 96.3% compared to *Bt*E264, respectively. We conducted PCR analyses on colonies from 2002721643 and 2002721723. Junction PCR products of the expected 2.2 kb size were obtained for 5/12 colonies from 2002721643 and 0/12 colonies from 2002721723, while all colonies yielded 1 kb PCR products when the control

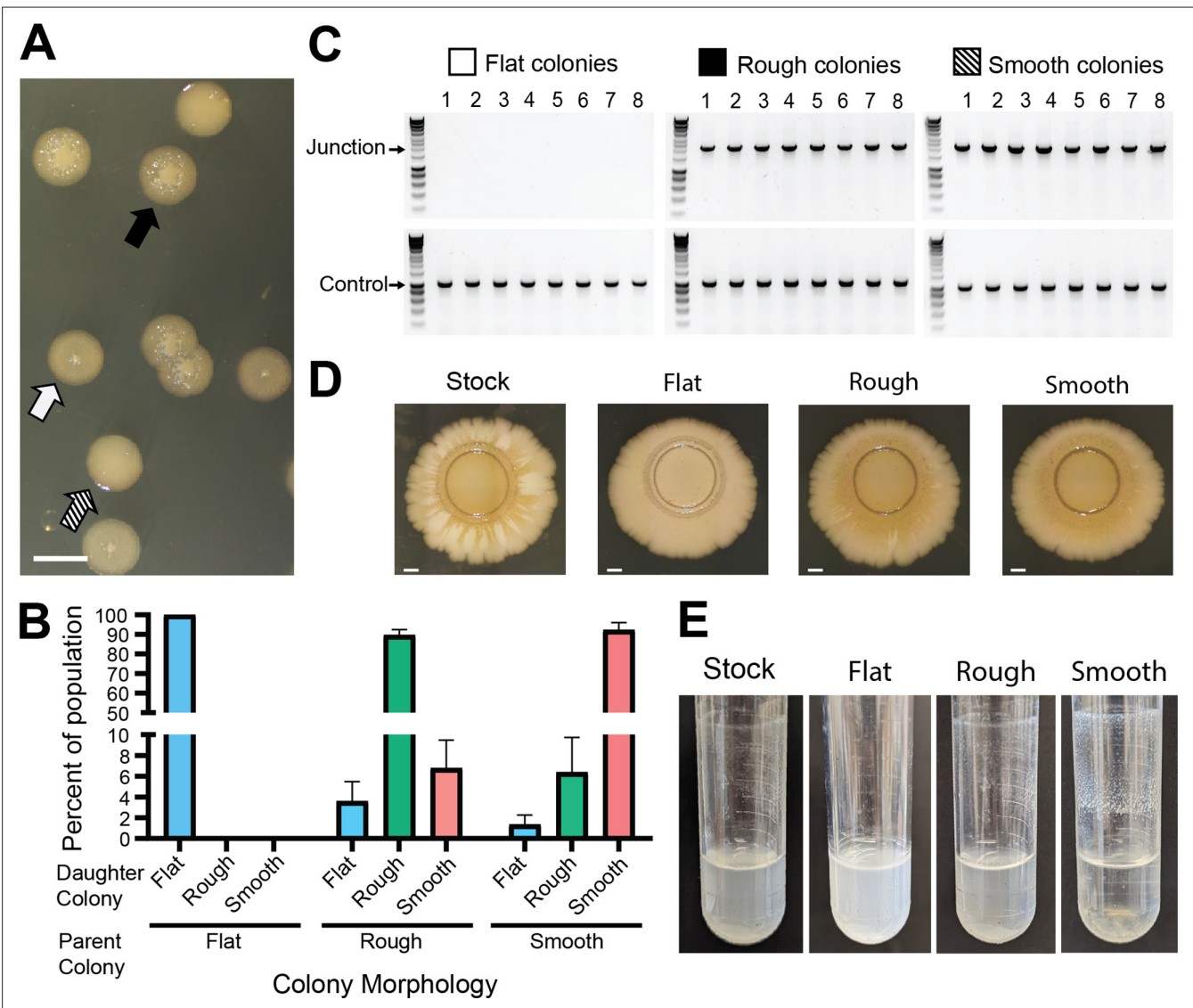

**Figure 2.** Genotypic heterogeneity correlates with phenotypic heterogeneity. (**A**) Image of the three *Bt*E264 colony phenotypes – flat (white arrow), raised rough (black arrow), and raised smooth (striped arrow) – observed on LSLB agar. Scale bar: 2 mm. (**B**) Colonies of each morphology were restreaked and the percentage of each morphology within the population of daughter colonies was calculated. (**C**) Eight colonies of each morphology were used as templates for PCR using Junction (Junc1 and Junc2) or Control (Ctrl1 and Ctrl2) primers. (**D**) Images of colony biofilms from the *Bt*E264 stock, flat colonies, rough raised colonies, and raised smooth colonies. Scale bar: 2 mm. Data are representative of three independent biological replicates. (**E**) Images of test tube aggregation seeded from the *Bt*E264 stock, flat colonies, raised rough colonies, and raised smooth colonies. Data are representative of three independent biological replicates.

The online version of this article includes the following source data for figure 2:

**Source data 1.** Uncropped DNA gel displaying junction and control PCRs of flat colonies that was used to make *Figure 2C*.

**Source data 2.** Uncropped DNA gel displaying junction and control PCRs of rough colonies that was used to make *Figure 2C*.

**Source data 3.** Uncropped DNA gel displaying junction and control PCRs of smooth colonies that was used to make *Figure 2C*.

primers were used (*Figure 1F, G*). These results suggest that *B. thailandensis* strains closely related to *Bt*E264 also exhibit variation in copy number of a similar region.

## Genotypic heterogeneity correlates with phenotypic heterogeneity

We noticed that although colonies from our *Bt*E264 stock appeared identical after 24 hr growth on LSLB agar, three distinct morphologies were apparent after 48 hr. Nearly half of the colonies were flat with slightly raised centers (i.e., umbonate) (*Figure 2A*, white arrow), the other half were raised and rough (*Figure 2A*, black arrow), and about 1/75 were raised and smooth (*Figure 2A*, striped arrow). Restreaking flat colonies yielded only daughter colonies with the same flat morphology (*Figure 2B*), while restreaking raised rough and raised smooth colonies yielded daughter colonies wherein 90% exhibited the parental colony morphology and the remainder exhibited other morphologies (*Figure 2B*). All raised colonies (rough and smooth), and no flat colonies, yielded 2.2 kb junction PCR products (*Figure 2C*), demonstrating a correlation between colony morphology and the presence of two or more copies of the 208.6 kb region.

We investigated duplication-dependent phenotypes and found that bacteria taken directly from our stock of *Bt*E264 displayed intermediate phenotypes compared with junction PCR positive and junction PCR negative bacteria. For example, the brownish-gold coloration developed after 28 days in colony biofilms established by bacteria from raised (junction PCR positive) colonies, but not in colony biofilms established by bacteria from flat (junction PCR negative) colonies, while colony biofilms established by bacteria taken directly from our frozen stock developed the brownish-gold coloration in the center and in alternating areas radiating outward from the original spot (*Figure 2D*). When grown in M63 medium on a roller, bacteria from raised colonies, but not flat colonies, aggregated and adhered to the walls of the test tube at the air–liquid interface (*Figure 2E*). Aggregation was apparent in the culture initiated with bacteria from our stock of *Bt*E264, but it was substantially less than that of the culture from raised colonies. These data further demonstrate the heterogeneity of our stock of *Bt*E264 and confirm previous observations that specific phenotypes correlate with the presence of two or more copies of the 208.6 kb region.

## The 208.6 kb region is present as directly oriented, tandem repeats in the chromosome of junction PCR positive bacteria

We previously concluded that the 208.6 kb region amplified as extrachromosomal, circular, DNA molecules (*Ocasio and Cotter, 2019*). However, subsequent genetic experiments investigating the requirements for the formation of the extrachromosomal circle led us to question that conclusion. To distinguish between extrachromosomal circles and tandem duplications in the chromosome (both of which would generate the same junction and hence yield the same junction PCR product), we engineered strains with unique I-*Sce*I restriction endonuclease sites located approximately 50 kb outside the boundaries of the 208.6 kb region (*Figure 3A*). Agarose plugs of cultures from flat, raised rough, and raised smooth colonies containing the I-*Sce*I sites were digested with I-*Sce*I, then the DNA was separated by pulsed-field gel electrophoresis (PFGE). A single band of ~310 kb was present in I-*Sce*I-digested DNA from flat colonies, indicating the presence of one copy of the 208.6 kb region in the chromosome (*Figure 3B*). I-*Sce*I digestion of DNA from raised rough colonies resulted in a prominent band of ~520 kb, and faint bands of ~310 and ~730 kb. This result indicates that a majority of the bacteria in the raised rough colony contain two tandem copies of the 208.6 kb region, and minor populations in the colony contain either a single copy (~310 kb band) of the region or three tandem copies (~730 kb band) of the region. DNA from the raised smooth colonies yielded a prominent ~730 kb band, with faint bands corresponding to ~520, ~310, and ~970 kb, indicating that a majority of the bacteria in raised smooth colonies contain three copies of the 208.6 kb region. No bands of ~210 kb, the size expected for an extrachromosomal circle, were detected in any sample. While junction PCR suggests that tandem repeats are oriented in a direct orientation, we wanted to further verify the directionality of the duplications. Plugs from flat, raised rough, and raised smooth colonies were digested with *Spe*I which is native to a locus ~143 kb within the duplicating region (*Figure 3—figure supplement 1A, B*). Directionality of the duplication will lead to differently sized fragments (~208 kb for a direct orientation and ~132 kb for an inverted orientation). We observed ~208 kb fragments indicating that the region amplifies as directly oriented copies. Our data demonstrate that the 208.6 kb region is present in directly oriented tandem copies in the chromosome and not extrachromosomal

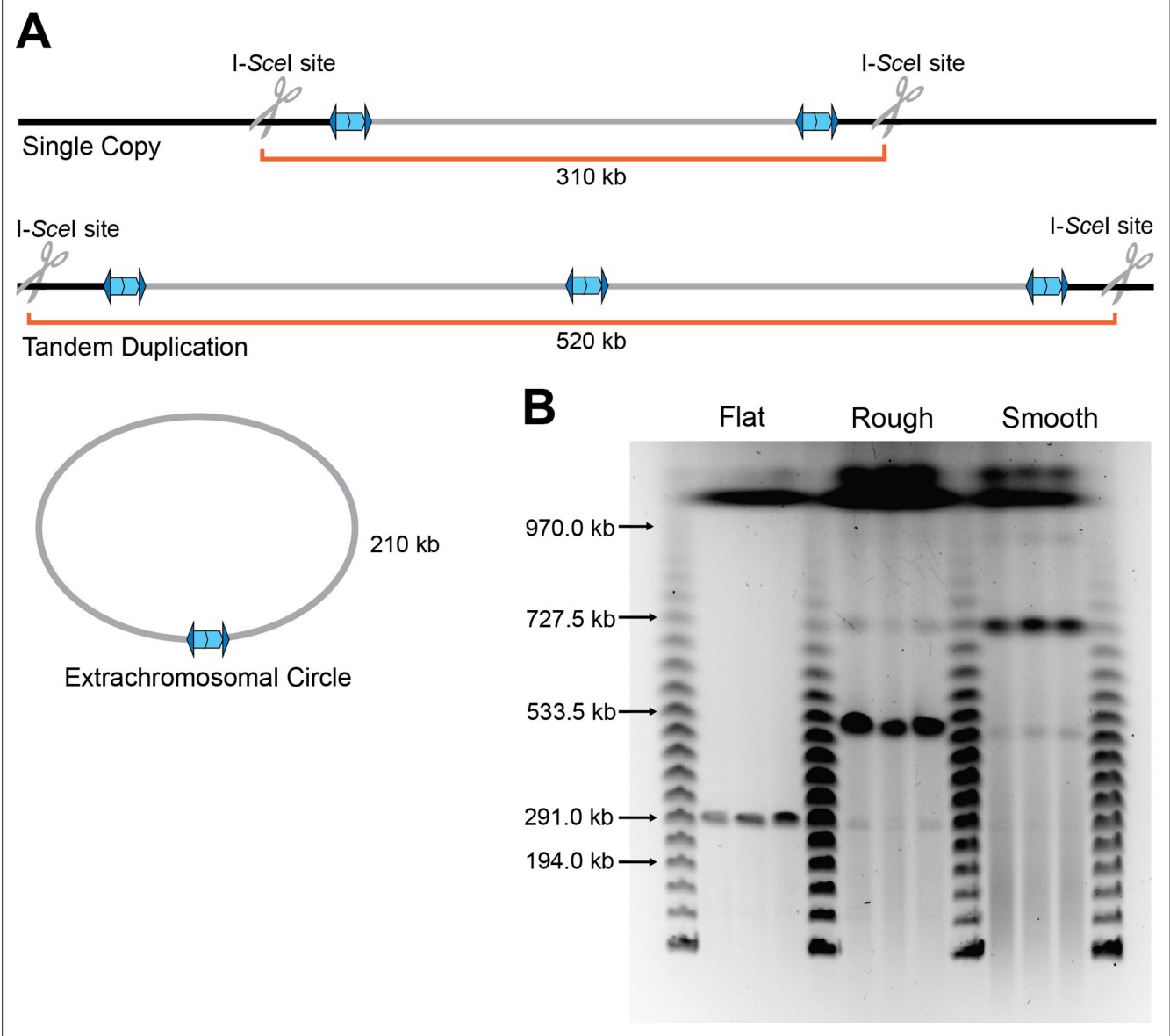

**Figure 3.** The duplication is present as a directly oriented tandem repeat of the 208.6 kb region in the chromosome. (**A**) Graphical representation of *Bt*E264 with engineered unique I-*Sce*I cut sites ~50 kb outside of the 208.6 kb duplicating region creating differently sized fragments depending on the form of the duplication. (**B**) Pulsed-field gel electrophoresis separated duplicating region fragments, post I-*Sce*I digestion, from three flat, raised rough, and raised smooth colonies each alongside lambda ladder. Data are representative of two independent biological replicates.

The online version of this article includes the following source data and figure supplement(s) for figure 3:

**Source data 1.** Uncropped pulsed-field gel electrophoresis (PFGE) DNA gel that was used to make *Figure 3B*.

**Figure supplement 1.** Tandem duplications form in a direct orientation.

**Figure supplement 1—source data 1.** Uncropped pulsed-field gel electrophoresis (PFGE) DNA gel that was used to make *Figure 3—figure supplement 1B*.

circular DNA molecules. These results suggest that amplification and resolution of the 208.6 kb region within a growing population of bacteria is dynamic. For simplicity, we will henceforth refer to junction PCR positive bacteria (which contain two or more copies of the 208.6 kb region) as Dup[+] and junction PCR negative bacteria (which contain a single copy of the region) as Dup[−].

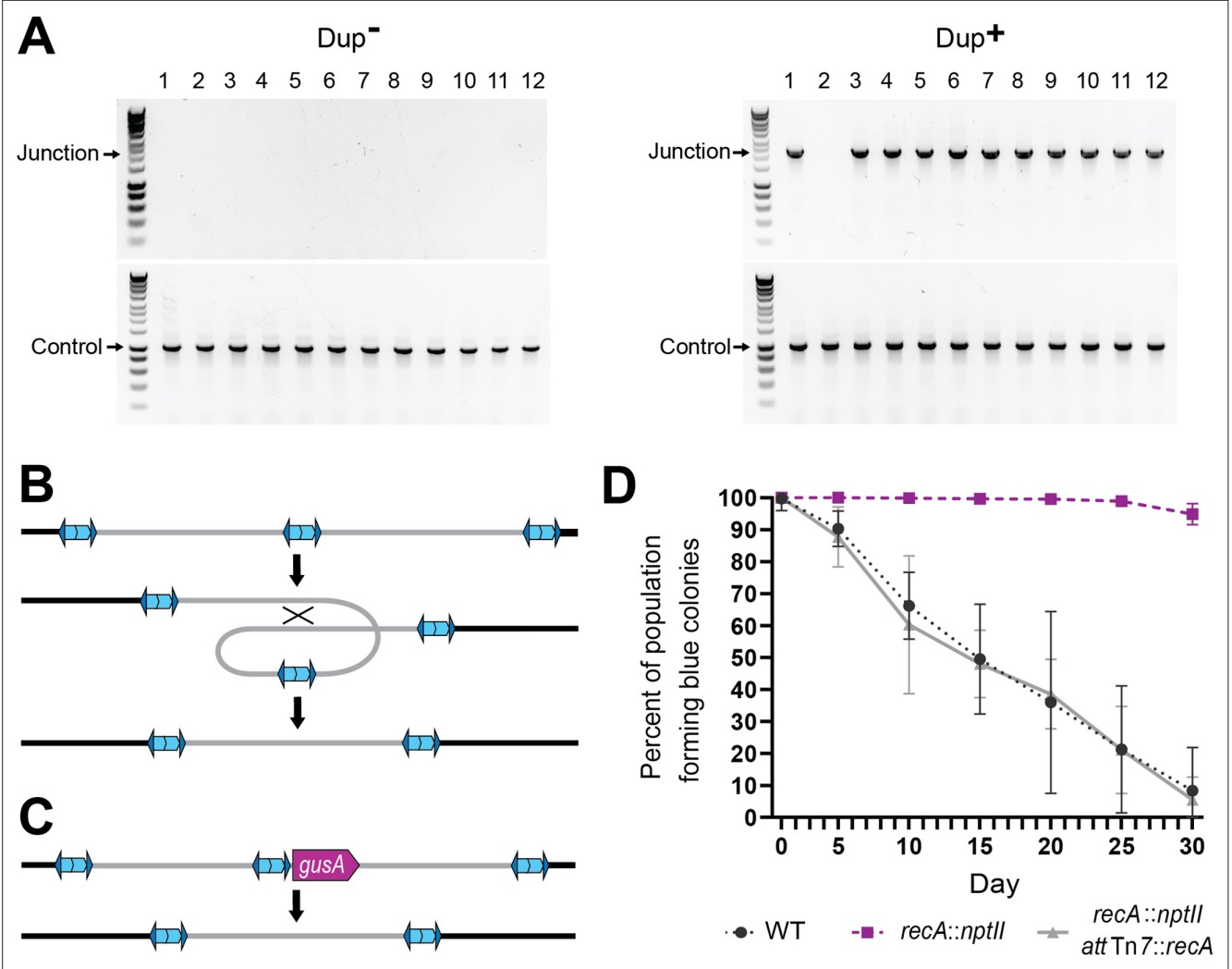

**Figure 4.** Resolution of tandem duplications occurs through RecA-dependent homologous recombination. (**A**) Twelve daughter colonies from a Dup⁺ and Dup⁻ colony were used as templates for PCR using either Junction (Junc1 and Junc2) or Control (Ctrl1 and Ctrl2) primers. (**B**) Model for tandem resolution by recombination between homologous sequences in the duplicated region. (**C**) Schematic of the *gusA* reporter strain used to visualize colonies that have resolved the tandem duplication. (**D**) Percentage of WT, *recA::nptII*, and *recA::nptII att*Tn7::*recA* colonies containing *gusA* throughout daily liquid subculturing. Mean and standard deviation plotted. From day 5 onward, the percent of blue colonies are significantly different between strains with and without functional *recA*, p values <0.01 at days 5–30 as analyzed by Kruskal–Wallis test.

The online version of this article includes the following source data for figure 4:

**Source data 1.** Uncropped DNA gel displaying junction PCR of colonies from Dup⁺ parent colonies that was used to make *Figure 4A*.

**Source data 2.** Uncropped DNA gel displaying junction PCR of colonies from Dup⁻ parent colonies that was used to make *Figure 4A*.

**Source data 3.** Uncropped DNA gel displaying control PCR of colonies from Dup⁺ and Dup⁻ parent colonies that was used to make *Figure 4A*.

## Resolution of tandem duplications occurs through RecA-dependent homologous recombination

When we restreaked Dup⁻ colonies on LSLB agar, we found that of hundreds of daughter colonies tested, none yielded a junction PCR product, indicating the presence of only a single copy of the 208.6 kb region (*Figure 4A*). When we restreaked Dup⁺ colonies on LSLB agar, ~95% of the daughter colonies yielded a positive junction PCR product, while ~5% of the daughter colonies failed to yield a junction PCR product, indicating that the tandem duplication had resolved to a single copy in ~5% of the bacteria in each Dup⁺ colony. These results indicate that resolution of tandem copies of the region to a single copy occurs during growth as a single colony on LSLB agar.

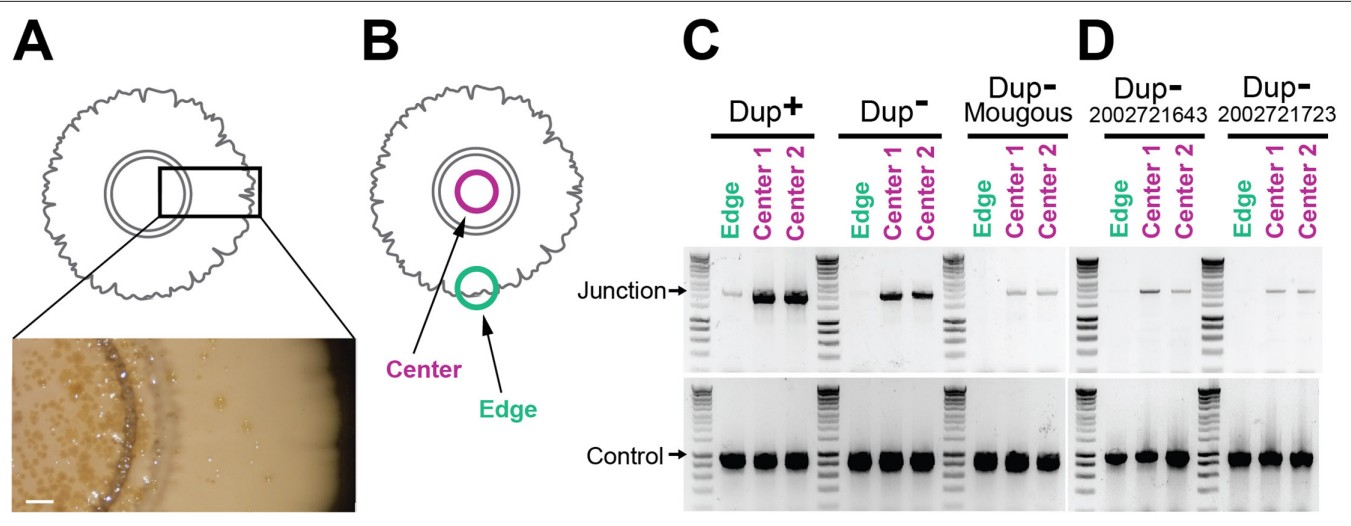

**Figure 5.** Duplications are enriched for in the center of colony biofilms. (**A**) Colony biofilm schematic and image depicting brownish-gold spots form primarily in the center of Dup⁻ colony biofilms. Scale bar: 2 mm. (**B**) Schematic demarcating the center and the edge portions of colony biofilms. The center and edge of colony biofilms from (**C**) wild-type Dup⁺, wild-type Dup⁻, and Mougous laboratory *Bt*E264 and (**D**) *B. thailandensis* 2002721643 and 2002721723 were used as templates for PCR using Junction (Junc1 and Junc2) or Control (Ctrl1 and Ctrl2) primers.

The online version of this article includes the following source data for figure 5:

**Source data 1.** Uncropped DNA gel displaying junction PCR from wild-type Dup⁺, wild-type Dup⁻, and Mougous Laboratory's wild-type strains that was used to make *Figure 5C*.

**Source data 2.** Uncropped DNA gel displaying control PCR from wild-type Dup⁺, wild-type Dup⁻, and Mougous Laboratory's wild-type strains that was used to make *Figure 5C*.

**Source data 3.** Uncropped DNA gel displaying junction PCR from strains 2002721643 and 2002721723 that was used to make *Figure 5C*.

**Source data 4.** Uncropped DNA gel displaying control PCR from strains 2002721643 and 2002721723 that was used to make *Figure 5C*.

We hypothesized that resolution of tandem copies of the 208.6 kb region to a single copy occurs through RecA-dependent homologous recombination, which could occur anywhere within the 208.6 kb of homologous sequences, resulting in a deletion of an entire copy of the region (*Figure 4B*). To test this hypothesis, we constructed a reporter strain in which we inserted *gusA* adjacent to the unique junction in a Dup⁺ strain (*Figure 4C*). The product of *gusA* hydrolyzes X-Gluc, releasing chloro-bromindigo and resulting in the formation of blue colonies. With *gusA* located near the center of the duplicated region, resolution of a tandem duplication to a single copy should, in the vast majority of cases, result in loss of *gusA* and hence the formation of white colonies on plates containing X-Gluc. We also constructed a *recA*-deficient derivative of this strain (Δ*recA::nptII*), and a derivative of the Δ*recA::nptII* strain in which *recA*, under the control of its native promoter, was inserted into both of the *att*Tn7 sites in the *Bt*E264 genome (Δ*recA::nptII att*Tn7::*recA*). We inoculated LSLB with the *gusA*-containing strains, and every 24 hr we collected aliquots for subculturing into fresh LSLB and for plating on X-Gluc-containing agar. The proportion of blue colonies obtained from our otherwise wild-type *gusA* insertion strain decreased from 100% on day 1 to ~10% by day 30 (*Figure 4D*). The proportion of blue colonies was maintained at nearly 100% over the 30-day course of the experiment in the *recA*-deficient strain, but the complemented strain mirrored wild-type and dropped to ~10% by day 30. These data indicate that resolution of tandem copies of the 208.6 kb region to a single copy during growth in LSLB occurs by *recA*-dependent homologous recombination.

## Duplication of the 208.6 kb region is enriched in the center of colony biofilms

We observed that when colony biofilms of Dup⁻ cells were grown at room temperature for an extended amount of time (>4 weeks), brownish-gold spots appeared in the center (*Figure 5A*). The coloration was reminiscent of that which appears throughout the middle of colony biofilms initiated with Dup⁺ bacteria. We hypothesized that these spots were clusters of cells in which the 208.6 kb region had

amplified. To attempt to speed up the process, we seeded colony biofilms with a 5× denser culture of cells. We established colony biofilms with $3.6 \times 10^7$ cfu instead of $7.2 \times 10^6$ cfu in a 20 µl droplet. After 7 days of growth, we picked bacteria from the center and edge of the colony biofilm for PCR analysis (*Figure 5B*). In colony biofilms established with Dup⁺ bacteria, junction PCR products were obtained from bacteria picked from both the center and the edge (*Figure 5C*). In Dup⁻ colony biofilms, junction PCR products were obtained from bacteria picked from the center, but not the edge. We also tested colony biofilms established with Dup⁻ 2002721643, 2002721723, and *Bt*E264 from the Mougous laboratory and similarly observed that Junction PCR products were obtained from bacteria picked from the center, but not the edge, of the colony biofilms (*Figure 5D*). These results demonstrate that Dup⁻ *Bt*E264 bacteria from our laboratory and the Mougous laboratory, and strains 2002721643 and 2002721723, are capable of duplicating the 208.6 kb region and suggest that either duplication occurs more frequently during growth in the center of a colony biofilm, or that duplication of the region confers a growth or survival advantage to bacteria growing in the center of a dense colony biofilm.

## BcpAIOB activity is not involved in amplification of the 208.6 kb region

We previously concluded that the BcpAIOB contact-dependent inhibition system was required for amplification of the 208.6 kb region, and that the amplification occurred by the production of extra-chromosomal, circular, DNA molecules (*Ocasio and Cotter, 2019*). Having demonstrated that the 208.6 kb region in junction PCR positive wild-type E264 bacteria are present as tandem copies in the chromosome, we re-evaluated the form of the DNA amplification in *bcpAIOB* mutants. We introduced I-*Sce*I sites into the previously constructed Δ*bcpAIOB* and P$_{S12}$-*bcpAIOB* strains, which are Dup⁻ and Dup⁺, respectively, digested chromosomal DNA with I-*Sce*I, and examined the digested DNA by PFGE (*Anderson et al., 2012*). The digested DNA from the Δ*bcpAIOB* strain contained a fragment of ~310 kb, corresponding to a single copy of the region, while digested DNA from the P$_{S12}$-*bcpAIOB* strain contained a prominent ~520 kb band, corresponding to the size expected for a tandem duplication of the 208.6 kb region, and faint bands at ~310 and ~730 kb, corresponding to fragments containing one and three copies of the region, respectively (*Figure 6—figure supplement 1*). These results indicate that the presence of the unique junction sequence in our *bcpAIOB* mutant strains correlates with multiple or single chromosomal copies of the 208.6 kb region in the chromosome, as it does for the flat and raised colonies from our frozen stock of *Bt*E264.

We were not aware of the heterogeneity in our wild-type *Bt*E264 stock when constructing the original Δ*bcpAIOB* and P$_{S12}$-*bcpAIOB* strains, causing us to generate strains with inconsistent copy number of the 208.6 kb region. We therefore re-evaluated whether a correlation between BcpAIOB activity and copy number of the 208.6 kb region (and duplication-dependent phenotypes) actually exists by constructing new Δ*bcpAIOB* and P$_{S12}$-*bcpAIOB* strains. Introduction of the S12 promoter upstream of the *bcpAIOB* genes in a Dup⁻ strain resulted in a Dup⁻ strain that lacked duplication-dependent phenotypes (Figure S2B), and deletion of both copies of *bcpAIOB* in a Dup⁺ strain (Δ*bcpAIOB::ble* Δ*bcpAIOB::nptII*) resulted in a Dup⁺ strain that displayed duplication-dependent phenotypes (Figure S2C). We also grew Dup⁻ strains with the *bcpAIOB* mutations as colony biofilms and conducted PCR analyses. Junction PCR products were obtained from bacteria picked from the center of all colony biofilms, including the Δ*bcpAIOB* strain (*Figure 6A*). Together, these results indicate that duplication-dependent phenotypes correlate with the presence of two or more copies of the 208.6 kb region and not with BcpAIOB activity (even indirectly), and they show definitively that amplification of the 208.6 kb region does not require BcpAIOB activity.

## ISα and ISβ transposase activity is not required for amplification of the 208.6 kb region

We also previously concluded that ISβ was required for amplification of the 208.6 kb region, based on data from strains constructed before our awareness of copy number variability of the 208.6 kb region in our stock of *Bt*E264. To re-evaluate the roles of ISα and ISβ in amplification of the 208.6 kb region, we separately replaced each IS element, including the inverted repeats that flank the transposase-encoding *orfAB* genes, with *nptII* (confers kanamycin resistance) in Dup⁻ bacteria, grew the two strains as colony biofilms, and conducted PCR analyses. While PCR products were obtained from bacteria picked from the centers and edges of the colony biofilms when control primers were used, no junction

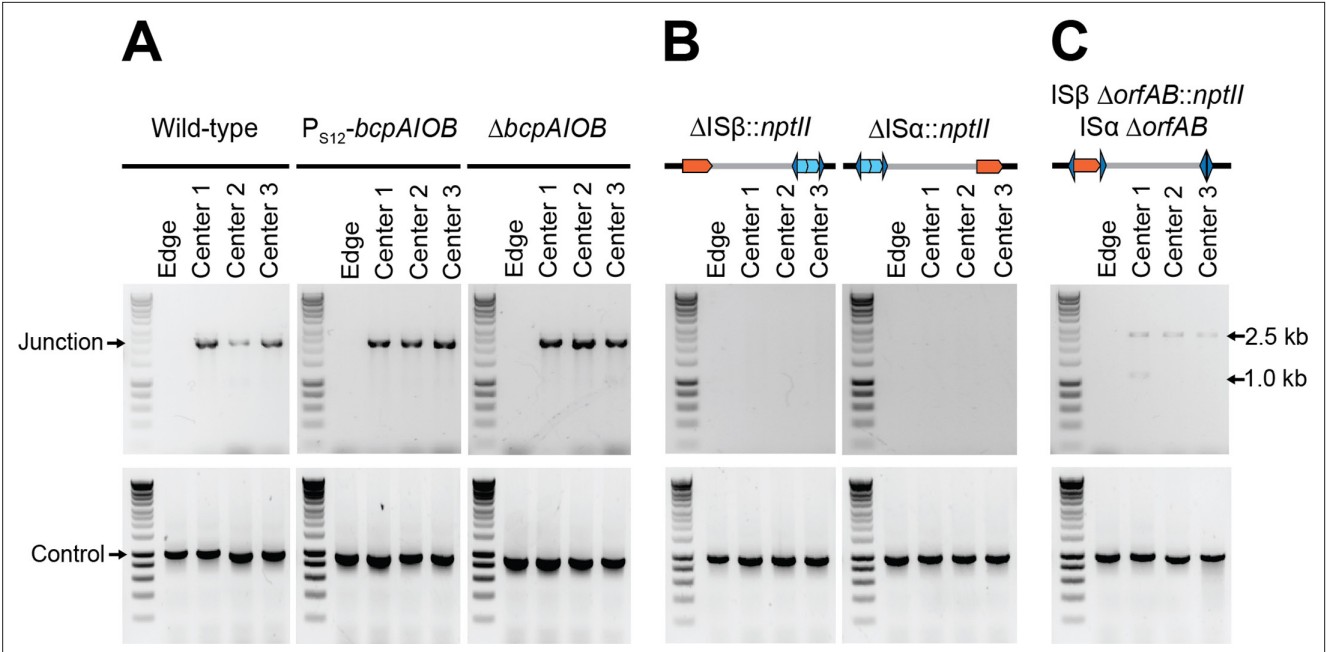

**Figure 6.** Contact-dependent growth inhibition (CDI) system-encoding *bcpAIOB* and transposase-encoding *orfAB* genes are not involved in duplication of the 208.6 kb region. (**A**) The center and edge of colony biofilms from wild-type Dup⁺, wild-type Dup⁻, Dup⁺ P_{S12}-*bcpAIOB*, and Dup⁻ Δ*bcpAIOB* were used as templates for PCR using Junction (Junc1 and Junc2) or Control (Ctrl1 and Ctrl2) primers. (**B**) The center and edge of colony biofilms from ΔISβ::*nptII* and ΔISα::*nptII* were used as templates for PCR using Junction (Junc1 and Junc2) or Control (Ctrl1 and Ctrl2) primers. (**C**) The center and edge of colony biofilms from ISβ Δ*orfAB::nptII* ISα Δ*orfAB* were used as a template for PCR using Junction (Junc1 and Junc2) or Control (Ctrl1 and Ctrl2) primers.

The online version of this article includes the following source data and figure supplement(s) for figure 6:

**Source data 1.** Uncropped DNA gel displaying junction PCR from wild-type, P_{S12}-*bcpAIOB*, Δ*bcpAIOB*, and ISβ Δ*orfAB::nptII* ISα Δ*orfAB* strains that was used to make *Figure 6A, C*.

**Source data 2.** Uncropped DNA gel displaying junction PCR from ΔISβ::*nptII* and ΔISα::*nptII* strains that was used to make *Figure 6B*.

**Source data 3.** Uncropped DNA gel displaying control PCR from wild-type, P_{S12}-*bcpAIOB*, and Δ*bcpAIOB* strains that was used to make *Figure 6A*.

**Source data 4.** Uncropped DNA gel displaying control PCR from ΔISβ::*nptII*, ΔISα::*nptII*, and ISβ Δ*orfAB::nptII* ISα Δ*orfAB* strains that was used to make *Figure 6B, C*.

**Figure supplement 1.** Phenotypes are duplication dependent and not *bcpAIOB* dependent.

**Figure supplement 1—source data 1.** Uncropped pulsed-field gel electrophoresis (PFGE) DNA gel that was used to make *Figure 6—figure supplement 1A*.

PCR products were obtained from the ISα and ISβ mutants (*Figure 6B*). To determine specifically if the transposase encoded by either ISα or ISβ is required for amplification of the region, we constructed a strain (ISα Δ*orfAB* ISβ Δ*orfAB::nptII*) that lacked *orfAB* from both ISα and ISβ while leaving the flanking inverted repeats for both elements intact, and grew this strain as a colony biofilm. In all five colony biofilms tested, junction PCR yielded a ~2.6 kb product that is of the predicted size for a junction that contains the *nptII* gene, and DNA sequence analysis confirmed that this fragment contains the *nptII* gene flanked by sequences representing the 5' and 3' ends of the duplicated region, matching the ISβ loci (*Figure 6C*). In one of the colony biofilms (center 1), an additional junction PCR product of ~1.1 kb was obtained, matching the ISα locus. DNA sequence analysis confirmed that this PCR product contained a junction containing the inverted repeats and not the *nptII* gene. These data show that the transposases encoded by ISα and ISβ are not required for amplification of the 208.6 kb region, but the inverted repeats are apparently necessary and sufficient for amplification of the 208.6 kb region.

## Amplification of the 208.6 kb region occurs by RecA-mediated recombination between homologous sequences

Because transposases of the same family have been shown to only function in cis (*Chandler et al., 2015*), we thought it unlikely that inverted repeat-mediated duplication of the region was catalyzed by one (or more) of the four other IS2-like transposases encoded in the genome. Instead, we hypothesized that duplication, like resolution, occurs through RecA-mediated recombination between homologous sequences. To test this hypothesis, we grew colony biofilms from wild-type, *recA*-deficient

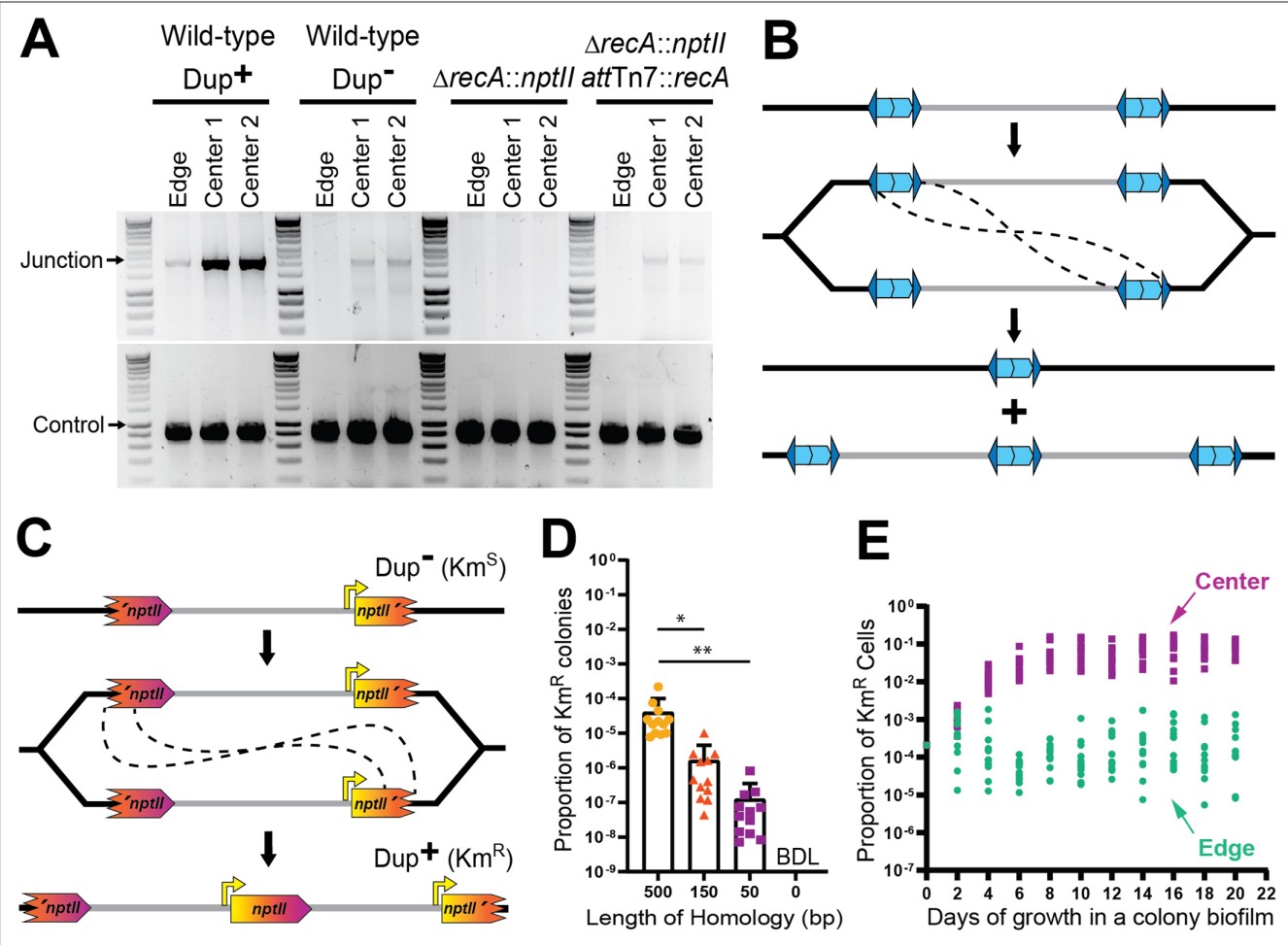

**Figure 7.** Duplications are formed through RecA-mediated recombination between homologous IS elements. (**A**) The center and edge of colony biofilms from WT Dup⁺, WT Dup⁻, Δ*recA*, and Δ*recA att*Tn7::*recA* were used as templates for PCR using Junction (Junc1 and Junc2) or Control (Ctrl1 and Ctrl2) primers. (**B**) Model for homologous recombination during replication resulting in two different genotypes within the daughter cells. One daughter cell contains a suspected fatal deletion of the 208.6 kb region while the other cell contains a tandem duplication of the 208.6 kb region. (**C**) Schematic of the fragmented *nptII* reporter system. Through recombination between homologous *nptII* sequences, a Km$^S$ cell can give rise to a Km$^R$ daughter cell with an intact *nptII* present at the junction. Orientation of the chromosome has been reversed for clarity and consistency. (**D**) Km$^R$ resistance frequency among fragmented *nptII* reporter strains with various amounts of homology (500, 150, 50, or 0 bp) following culture in LSLB broth. Mean and standard deviation indicated. 'BDL' indicates below detection limit. * and ** denote p values <0.05 and <0.01, respectively, as calculated by an analysis of variance (ANOVA). (**E**) Using the fragmented *nptII* reporter strain, calculated proportion of Km$^R$ cells from the center and edge of colony biofilms over time. From day 4 onwards, the proportion of Km$^R$ cells from the center and edge of the colony biofilm are significantly different; p values <0.0001 at days 4–20 as calculated by Mann–Whitney *U*.

The online version of this article includes the following source data for figure 7:

**Source data 1.** Uncropped DNA gel displaying junction PCR from wild-type Dup⁺, wild-type Dup⁻, Δ*recA::nptII*, and Δ*recA::nptII att*Tn7::*recA* strains that was used to make **Figure 7A**.

**Source data 2.** Uncropped DNA gel displaying control PCR from wild-type Dup⁺, wild-type Dup⁻, Δ*recA::nptII*, and Δ*recA::nptII att*Tn7::*recA* strains that was used to make **Figure 7A**.

(Δ*recA::nptII*), and *recA* complemented (Δ*recA::nptII att*Tn7::*recA*) strains and conducted PCR analyses. No junction PCR products were produced from *recA*-deficient bacteria, while junction PCR products were obtained from the wild-type and *recA* complemented strains (**Figure 7A**). Amplification of the 208.6 kb region, therefore, requires *recA*.

These data support the hypothesis that amplification of the 208.6 kb region occurs by RecA-dependent recombination between the 1336 bp of homology shared by ISα and ISβ (**Figure 7B**). Homologous recombination can occur during DNA replication, through either unequal crossing over or during the repair of a collapsed replication fork to create DNA duplications (**Hastings et al., 2009**; **Bzymek and Lovett, 2001**). While replication fork repair leads to a deletion or duplication of the DNA region between the homologous sequences, unequal crossing over will generate one daughter chromosome with a duplication and one daughter chromosome with a deletion of those sequences (**Hastings et al., 2009**; **Bzymek and Lovett, 2001**). If at least one essential gene is present within the region, the daughter cell containing the chromosome with the deletion will be non-viable. To further evaluate if RecA-dependent recombination is duplicating the region, we developed a 'fragmented reporter system'. We replaced ISβ with a 5′ portion of *nptII* and ISα with a 3′ portion of *nptII* (**Figure 7C**). The amount of overlap (in the center of *nptII*) in the strains varied between 500 and 0 bp. Recombination between the overlapping homologous sequences will result in the formation of an intact *nptII* gene, which will confer resistance to kanamycin (Km$^R$). The Km$^R$ cfu and total cfu were obtained after overnight growth in LSLB and the proportion of Km$^R$ colonies was calculated (Km$^R$ cfu/total cfu = proportion Km$^R$ colonies) as $10^{-4}$, $10^{-6}$, and $10^{-7}$ for the strains with 500, 150, and 50 bp of homology, respectively (**Figure 7D**). The strain lacking homology produced no Km$^R$ colonies. These data indicate that duplication of the region occurs by RecA-dependent recombination between homologous sequences, independent of the specific sequence. Thus, the inverted repeats of ISα and ISβ are not required for duplication of the region, unless they are the only source of DNA sequence homology. And because homologous recombination regularly occurs at a low level, these data also suggest that wild-type *Bt*E264 exists as a heterogeneous population.

We can also use the fragmented reporter system to quantify population dynamics. When the fragmented *nptII* reporter strain with 500 bp of homology was grown as a colony biofilm, we observed that after 4 days of growth, the proportion of Km$^R$ bacteria from the center of the colony biofilm was consistently greater than the proportion of Km$^R$ bacteria from the edge (**Figure 7E**). These results further indicate that either duplication of the 208.6 kb region occurs more frequently at the center of the colony biofilm or, more likely, that bacteria with two or more copies of the region have a growth or survival advantage in the center of the colony biofilm.

## Amplification of the 208.6 kb region greatly enhances static biofilm development

Our data suggest a correlation between amplification of the 208.6 kb region and biofilm formation. We therefore developed a fragmented *gfp* (encoding green fluorescent protein) reporter system (**Figure 8A**) and compared strains for their ability to form biofilms using a standard submerged, static biofilm assay on glass coverslips in M63 medium. Each strain contains *rfp*, encoding red fluorescent protein, at a location outside the 208.6 kb region. Biofilms were initiated with Dup$^+$ or Dup$^-$ bacteria. In addition, to determine if the ability to amplify the 208.6 kb region is required for biofilm development, we used a Dup$^-$-locked strain that has an intact ISβ at one boundary of the region and the 5′ fragment of *gfp* replacing ISα at the other, and hence cannot duplicate the region and form GFP$^+$ bacteria. By 24 hr, the Dup$^+$ bacteria had formed biofilms with characteristic pillars of bacteria, while the Dup$^-$ and Dup$^-$-locked strains formed only a few, small aggregates of bacteria. By 72 hr, biofilms formed by Dup$^+$ bacteria were significantly taller and covering significantly more surface area than biofilms formed by Dup$^-$ and Dup$^-$-locked bacteria (**Figure 8B–D**). These data show that the presence of two or more copies of the 208.6 kb region greatly enhances biofilm formation by *Bt*E264. Several clumps of RFP$^+$GFP$^-$ cells were present in the biofilms formed by Dup$^+$ bacteria, indicating that resolution to a single copy of the 208.6 kb region can occur during biofilm development, and that bacteria containing only a single copy of the region can be maintained in the biofilm, at least transiently. By contrast, RFP$^+$GFP$^+$ bacteria were rarely observed in the biofilms formed by Dup$^-$ bacteria. These results indicate that in this static biofilm experimental condition, Dup$^+$ bacteria are not enriched within the population.

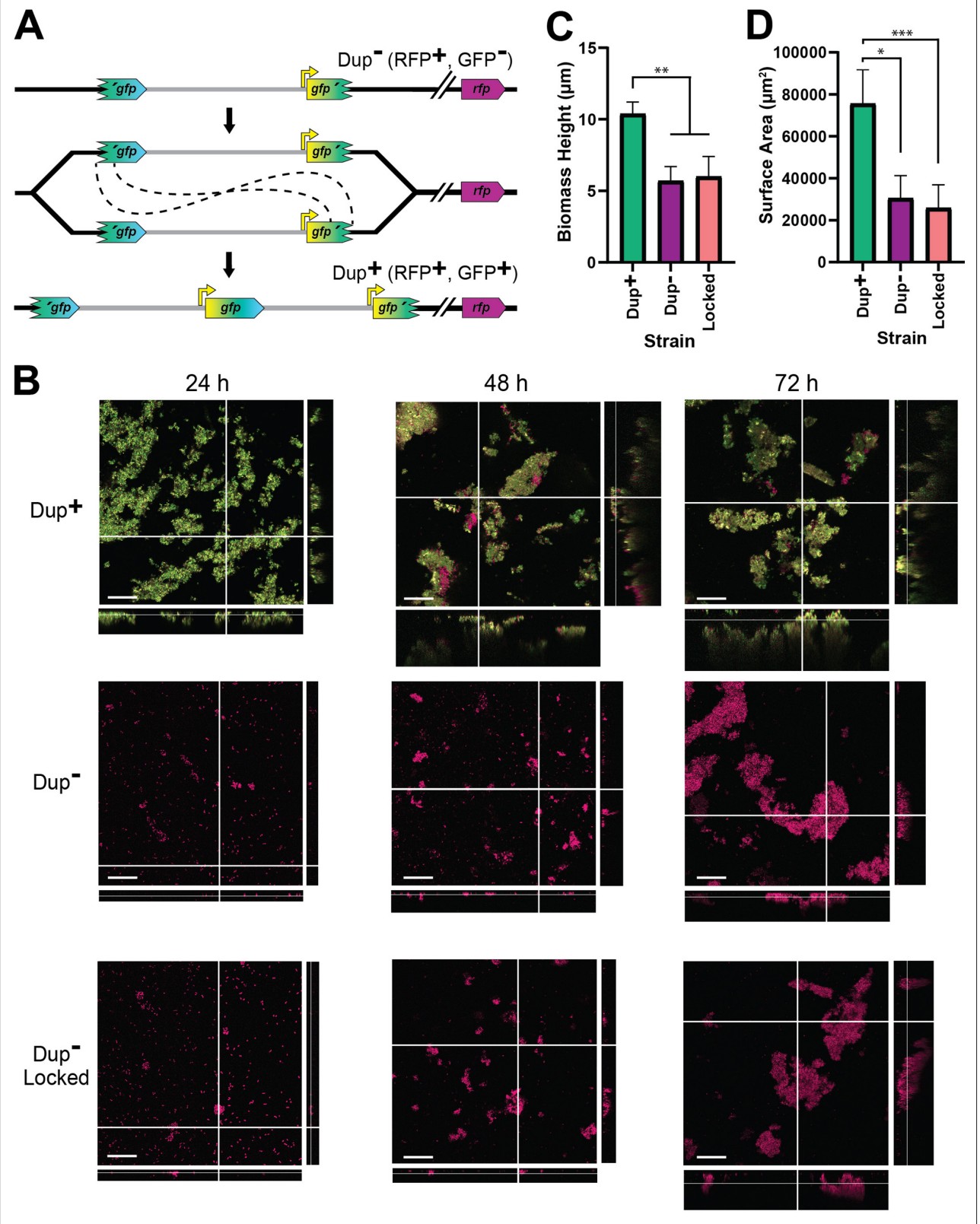

**Figure 8.** Duplication of the region greatly enhances biofilm development. (**A**) Schematic of the fragmented *gfp* reporter system. Through recombination between homologous *gfp* sequences, a GFP⁻ cell can give rise to a GFP⁺ daughter cell with an intact *gfp* present at the junction. (**B**) Z-stack confocal microscope images of biofilms from Dup⁺, Dup⁻, and locked fragmented *gfp* reporter strains at 24, 48, and 72 hr of growth. Scale bar: 30 μm. Average biofilm height (**C**) and surface area covered (**D**) from Dup⁺, Dup⁻, and locked fragmented *gfp* reporter strains at 72 hr. Columns indicate

*Figure 8 continued on next page*

*Figure 8 continued*

sample means and error bars correspond to standard deviation. *, **, and *** denote p values <0.05, <0.01, and <0.001, respectively, as produced by Kruskal–Wallis test. Data are representative of eight independent biological replicates.

## Amplification of the 208.6 kb region in *Bt*E264 provides a selective advantage during dynamic biofilm development

We next investigated the contribution of the 208.6 kb region to biofilm formation under more dynamic conditions using a fragmented *gusA* reporter system (*Figure 9A*). The fragmented *gusA* reporter strain was chosen for two reasons: (1) it allows us to screen, rather select than for cells with a duplication and (2) the *gusA* gene is long enough to create a fragmented reporter system with the same homologous sequence length provided by ISα and ISβ in wild-type cells (1336 bp). We inoculated M63 medium with Dup⁺ bacteria, incubated the cultures on a roller overnight, and maintained selection for biofilm growth by decanting the spent medium from the tube, sampling a portion of the bacteria at the air–liquid interface for plating on X-Gluc-containing plates, and then adding fresh medium to the same tube and repeating the process daily for 7 days. We also maintained selection for planktonic growth by sampling an aliquot from the liquid phase of the overnight culture for plating and for subculturing into fresh medium daily for 7 days. We quantified the proportion of blue colonies present on X-Gluc-containing LSLB agar to determine the frequency of cells with a duplication of the 208.6 kb region. The biofilm remained 100% Dup⁺ for the duration of the experiment, while the proportion of GusA⁺ bacteria in the liquid medium dropped to ~81% by day 1, ~16% by day 2, and ~0% by day 3. These data demonstrate that cells that resolve the tandem duplication do not remain in the biofilm. Under these dynamic conditions, growth in a biofilm selects for maintenance of a duplication of the 208.6 kb region, and planktonic growth selects for bacteria with a single copy of the region.

We next compared dynamic biofilm formation in M63 medium by Dup⁺, Dup⁻, and Dup⁻-locked bacteria. The Dup⁻-locked strain has an intact ISβ at one boundary of the region and the 3′ fragment of *gusA* at the other. As with the experiment described above, spent medium was replaced each day with fresh medium. At day 2, there were clearly aggregates of bacteria along the walls of the test tube that had been inoculated with Dup⁺ bacteria and only a barely visible film in the other cultures (*Figure 9C*). By day 4, however, cultures initiated with Dup⁺ and Dup⁻ bacteria had both formed similar thick, opaque biofilms at the air–liquid interface, while the tube inoculated with Dup⁻-locked bacteria still had only a barely visible film (*Figure 9C*). By day 6, a biofilm was present even in the Dup⁻-locked culture (*Figure 9C*). The time to visible biofilm formation is graphed in *Figure 9D*. These data support the conclusion that Dup⁺ bacteria have a substantial advantage during biofilm formation. They also indicate, however, that even bacteria that cannot duplicate the 208.6 kb region can ultimately form biofilms, albeit much less efficiently than bacteria that can duplicate the region.

Because a strain capable of duplicating the 208.6 kb region constructed biofilms more rapidly than a strain that could not, we hypothesized that cells in the biofilm contained a duplication. To test this, we repeated the biofilm and planktonic growth selection experiment using a Dup⁻ fragmented *gusA* reporter strain. Over the course of the experiment, the frequency of blue colonies from the broth culture remained around 0%, indicating that amplification of the 208.6 kb region in planktonically growing bacteria occurs at a very low frequency. Meanwhile, the proportion of blue colonies formed by bacteria recovered from the walls of the test tube increased to ~50% within 2 days, indicating a dramatic enrichment for Dup⁺ bacteria in the biofilm (*Figure 9E*).

## Discussion

We have discovered a phase variation system that allows *B. thailandensis* to adapt quickly to environments that demand different (planktonic versus biofilm) lifestyles (*Figure 10*). We showed that a pair of IS*2*-like elements bounding a region containing 157 genes provide DNA sequence homology for RecA-mediated homologous recombination, resulting in the formation of tandem, directly repeated copies of the region in the chromosome. Resolution of tandem copies to a single copy also occurs by RecA-mediated homologous recombination. We showed that the presence of two or more copies of the 208.6 kb region promotes biofilm formation, while a single copy is advantageous for planktonic growth. Because amplification and resolution of the region occur at low and presumably constant (yet different) rates, all growing populations of *Bt*E264 are heterogeneous with regard to region copy

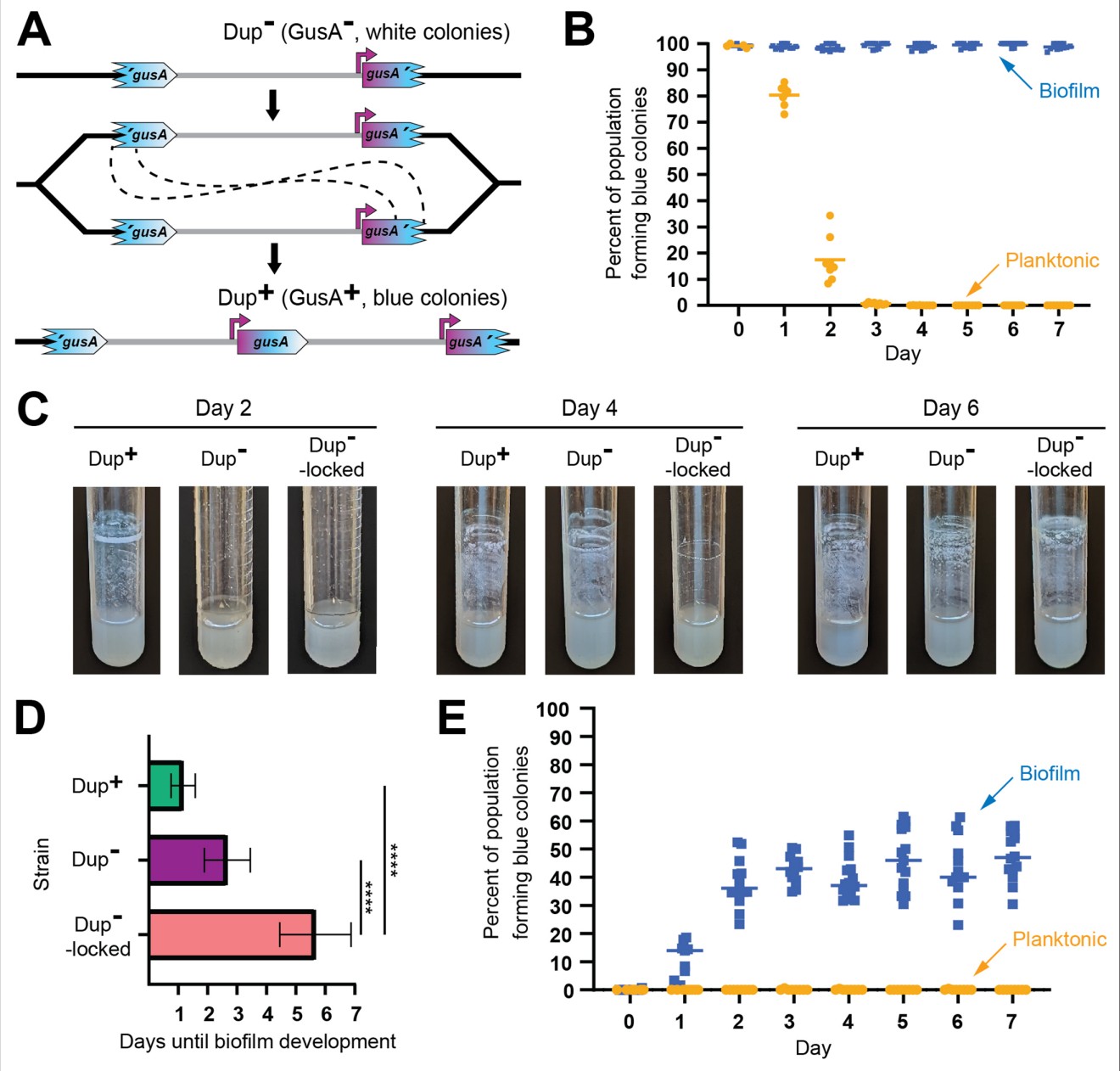

**Figure 9.** Planktonic growth enriches for a single copies of the 208.6 kb region while growth in a biofilm enriches for duplications. (**A**) Schematic of the fragmented *gusA* reporter system. Through recombination between homologous *gusA* sequences, a GusA⁻ cell can give rise to a GusA⁺ daughter cell with an intact *gusA* present at the junction. Orientation of the chromosome has been reversed for clarity and consistency. (**B**) Dup⁺ fragmented *gusA* reporter strain biofilms and planktonic cells were subcultured for 7 days and the proportion of blue colonies formed on X-Gluc media was calculated daily. Bars indicate sample mean. From day 1 onwards, the percent of the population forming blue colonies was significantly different, p values <0.01 at time points 1–7 as analyzed by Mann–Whitney *U*. (**C**) Images of representative air–liquid interface biofilms of Dup⁺, Dup⁻, and locked Dup⁻ fragmented *gusA* strains at 2, 4, and 6 days of growth. (**D**) Days taken for Dup⁺, Dup⁻, and locked Dup⁻ fragmented gusA strains to form visible, opaque biofilms at the air–liquid interface. Bars plot sample means, and error bars correspond to standard deviation. **** denotes p values <0.0001. Data are from six independent biological replicates. (**E**) Dup⁻ fragmented *gusA* reporter strain biofilms and planktonic cells were subcultured for 7 days and the proportion of blue colonies formed on X-Gluc media was calculated daily. Bars indicate sample mean. Data are representative of three independent biological replicates. From day 1 onwards, the percent of the population forming blue colonies was significantly different, p values <0.01 at time points 1–7 as analyzed by Mann–Whitney *U*.

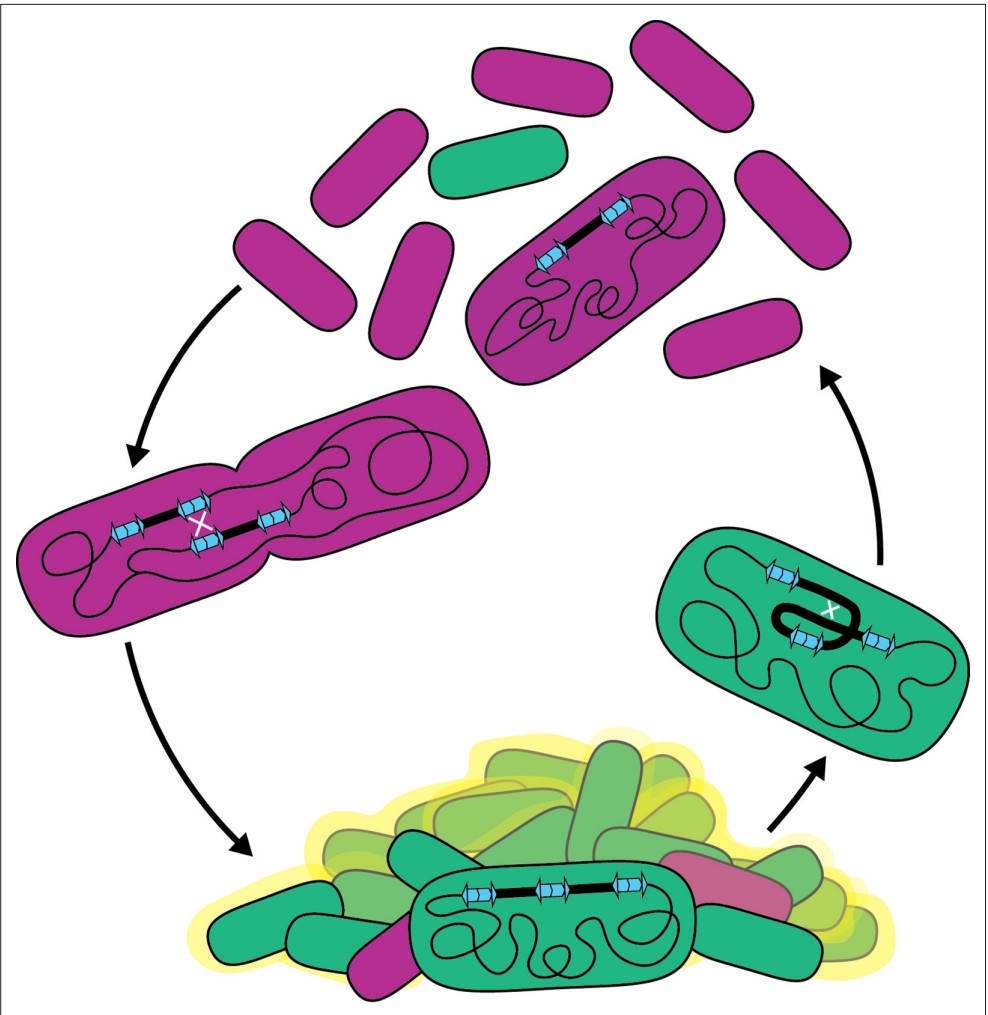

**Figure 10.** Model for the bet-hedging strategy in *B. thailandensis* strains containing ISα and ISβ. Recombination between homologous IS sequences forms heterogeneous populations that allows *B. thailandensis* to adapt to disparate growth environments.Bacterial biofilms are ubiquitous in both natural and manufactured environments. Because biofilms can contribute to human disease and can cause environmental harm, it is important to understand mechanisms underlying their formation and maintenance. Our data indicate that the presence of two or more copies of the 208.6 kb region promotes biofilm formation, suggesting that the responsible gene(s) act as an insufficient haplotype when the region is present in single copy. The 157 genes within the region are functionally diverse: they include gene, or gene clusters, predicted to encode defense systems, antimicrobial resistance, amino acid, carbohydrate, and lipid biosynthesis, transport, and metabolism pathways, and transcription regulators. Determining which gene(s), when present in two or more copies, facilitates biofilm formation will be our next goal. Genes encoding regulatory factors, such as predicted transcription regulators *lysR* and *luxR*, are likely candidates as alteration in copy number can push signaling pathways over a threshold, leading to large changes in gene expression for all genes within the regulon. Other candidate genes include those predicted to encode fimbria or a polysaccharide transport system, which may be directly involved in biofilm formation, but whether two or three copies of these genes is sufficient for biofilm formation seems unlikely, although we cannot rule it out. While biofilm-associated growth is a single condition benefited by duplication of the 208.6 kb region, we anticipate that duplications may benefit cells in other conditions. For example, we do not know the nature or function of the molecule(s) causing the gold-brown coloration that correlates with the presence of multiple copies of the region, but we speculate that it may be advantageous for the bacteria under conditions other than, or in addition to, biofilm formation and maintenance. The diversity of predicted gene functions within the duplicating region suggests that amplification has potential to benefit *Bt*E264 in multiple environments.

number, containing subpopulations that are optimized for either planktonic or sessile growth. This phenomenon, therefore, represents a bet-hedging strategy to improve population survival in fluctuating environments. Moreover, as it is likely that this phase variation system was formed by acquisition of the directly oriented IS2-like elements, we posit that it is an example of IS element-mediated evolution.

In the absence of a functional phase variation system (i.e., the Dup⁻-locked strain), BtE264 was still able to form a biofilm in the dynamic, test tube format, but it did so at a much slower rate compared to Dup⁻ bacteria. The delay is consistent with these bacteria acquiring spontaneous mutations, which occur at much lower frequencies than homologous recombination. Also consistent with these bacteria possessing spontaneous, biofilm-promoting mutations, the colonies recovered from those biofilms appeared mucoid, and preliminary data indicate they form biofilms at least as fast as Dup⁺ bacteria. Identifying the genetic alterations in these biofilm-forming, Dup⁻-locked strains will shed light on the molecular mechanism underlying biofilm formation and will help us determine why duplication of the region positively influences biofilm formation.

Copy number dynamics differed within populations growing in static versus dynamic biofilms. In dynamic (culture tube-associated) biofilms, duplication of the region provided a strong selective advantage; bacteria that lost the duplication were not maintained in the biofilm and dispersed into the planktonic population. By contrast, in static (submerged coverslip) biofilms, the selective advantage was not as apparent, even though Dup⁺ bacteria clearly formed more robust biofilms and did so more quickly than Dup⁻ bacteria. In static biofilms initiated with Dup⁺ bacteria, Dup⁻ bacteria arose and were maintained for at least for 3 days. Moreover, static biofilms initiated with Dup⁻ cells did not contain an appreciable number of Dup⁺ cells over the course of the experiment. We envisage four factors that may contribute to the difference in duplication dynamics between the two biofilm experiments.

1. Differences in culture conditions may impact the intensity of the selective pressure toward biofilm formation. While spent medium was replaced daily in both dynamic and static biofilms, the dynamic biofilms were constantly washed and agitated as the slanted culture tube rotated. We hypothesize that the constant washing of the dynamic biofilms removed less fit or less adherent Dup⁻ cells from the biofilm, whereas the static biofilm scenario simply did not provide the physical force required to remove those cells.
2. Duplication rates are impacted by the amount of homology provided by the fragmented gfp (500 bp) and gusA (1336 bp) reporter strains used in the static and dynamic biofilms, respectively. Our data indicate that duplications form more frequently when more homologous sequences bounding the region are longer (Figure 7D), potentially contributing to why we see more Dup⁺ strains in the fragmented gusA dynamic biofilms.
3. Lack of agitation, and hence lack of aeration, during static biofilm growth may increase generation time. Because recombination to duplicate the region occurs during DNA replication, duplication frequency is directly related to growth rate.
4. Differences in growth conditions may impact recA expression or RecA activity levels, thereby affecting duplication frequency.

Data acquired in this study refute some of our previous conclusions. Although we were previously unable to definitively determine the form of the 206.8 kb duplication, we concluded that the extra copies likely existed as extrachromosomal circular DNA molecules based on the fact that IS2 elements transpose using a copy-out-paste-in mechanism and the observation that the region mobilized to a locus in chromosome II when we attempted to delete the 206.8 kb region from chromosome I (Ocasio and Cotter, 2019). Our PFGE data prove that the region is present as tandem, directly oriented copies in the chromosome in Dup⁺ bacteria, and we have also now shown that the transposases encoded by ISα and ISβ are not required for amplification of the region. Extrachromosomal, circular copies of the region will occur, however, when homologous recombination resolves merodiploids (tandem duplications) to a single copy, and these circular DNA molecules can integrate into the chromosome by homologous recombination between the IS element in the circular molecule and any of the four or two IS2-like elements in chromosome I or II, respectively. Although such an event would be rare, our attempt to delete essential genes within the 206.8 kb region apparently provided the necessary selective pressure, explaining how the region moved to chromosome II in our previous study.

Our previous conclusion that BcpA activity is required for amplification of the 208.6 kb region was based on the fact that strains with mutations that inactivate BcpA (e.g., deletion of the bcpAIOB locus

or replacement of codons in *bcpA* required for BcpA catalytic activity) were junction PCR negative and did not display duplication-dependent phenotypes, and strains in which *bcpAIOB* was driven by the S12 promoter were strongly junction PCR positive and did display duplication-dependent phenotypes. It is now clear that when we constructed the BcpA-inactivating mutants (by either two-step allelic exchange or natural transformation and double homologous recombination) we chose Dup⁻ strains when we confirmed that the mutants contained the intended mutations and not the wild-type sequences. When we introduced the S12 promoter 5′ to *bcpAIOB* through natural transformation, however, we confirmed that the S12 promoter was present at the intended location, but had no knowledge at the time that there could be another copy of the locus in the chromosome, so, by chance, had chosen a Dup⁺ strain. Our data now clearly show that duplication-dependent phenotypes are due to the duplication of the region and not BcpAIOB activity.

Bacterial populations are often more genotypically heterogeneous than presumed. It is estimated that 10% of cells in a growing culture carry a duplication of some region of the chromosome (*Roth et al., 1996*). Unfortunately, standard alignment techniques call out SNPs, deletions, and inversions, but not duplications. Identifying duplications requires close attention to sequencing read abundance or long-read sequencing, and hence they are often missed, even when their existence is considered a possibility. Because duplications have the potential to alter a wide variety of phenotypes, they can lead to flawed conclusions when cells with and without DNA duplications are unknowingly compared. Discovering the 208.6 kb DNA duplication in *Bt*E264 allowed us to correct some of our previous conclusions, and also sheds light on other perplexing results. For example, a transposon screen of *Bt*E264 yielded strains with transposon insertions in the gene (*bcpI*) encoding the immunity protein for the BcpAIOB contact-dependent inhibition system, which should have been fatal (*Gallagher et al., 2013*). Such a mutation could be tolerated, however, if a second copy of the *bcpI* gene is present, and hence the Tn insertion into *bcpI* may have selected for Dup⁺ cells. Our data suggest that duplications should always be considered when mutants are obtained that contain mutations in genes predicted or suspected to be essential.

Transposable elements are facilitators of evolution. In the short term, insertion into coding sequences can impact gene expression (*Darmon and Leach, 2014*). In the long term, homologous transposable elements can act as substrates for DNA recombination that can alter genome architecture (*Vandecraen et al., 2017*). Notably, recombination between transposable elements facilitates genome reduction in host-evolved bacteria, which has been observed, for example, in the evolution of the equine-infecting *Burkholderia mallei* from the broader host range pathogen *Burkholderia pseudomallei* (*Losada et al., 2010*). We hypothesize that acquisition of ISα and ISβ at their current locations in *Bt*E264 represents an evolutionary step for *Bt*E264 – creating a bet-hedging phase variation system that allows for population survival when conditions oscillate between favoring planktonic and biofilm lifestyles. Here, we show that acquisition of a pair of transposable elements has constructed a phase variation system that allows *Bt*E264 to rapidly adapt to environments favoring planktonic or biofilm growth. Bioinformatic analysis of other *B. thailandensis* strains revealed that nearly all other related strains lack both ISα and ISβ. One exception that contains ISα but not ISβ and some adjacent sequences is a *B. thailandensis* strain that is unusually pathogenic called BPM (*Chang et al., 2017*). It is tempting to speculate that BPM represents an ancestral strain that has not acquired ISβ and the adjacent genes, although it may also represent a descendent that lost ISβ and the adjacent genes coincident with gaining virulence.

## Materials and methods

**Key resources table**

| Reagent type (species) or resource | Designation | Source or reference | Identifiers | Additional information |
|---|---|---|---|---|
| Gene (*Burkholderia thailandensis*) | Duplicating region | GCA_000012365.1 | LO74_RS28540– LO74_RS09425 | |
| Gene (*Burkholderia thailandensis*) | *recA* | GCA_000012365.1 | LO74_RS12520 | |

*Continued on next page*

*Continued*

| Reagent type (species) or resource | Designation | Source or reference | Identifiers | Additional information |
|---|---|---|---|---|
| Strain, strain background (*Burkholderia thailandensis*) | E264 | ATCC | 700388 | GCA_000012365.1 |
| Strain, strain background (*Burkholderia thailandensis*) | 2002721643 | USAMRIID | | GCA_000959425.1 |
| Strain, strain background (Burkholderia thailandensis) | 2002721723 | USAMRIID | | GCA_000567925.1 |
| Strain, strain background (*Escherichia coli*) | DH5α | GoldBio | CC-101-5x50 | Electrocompetent cells |
| Strain, strain background (*Escherichia coli*) | RHO3 | AddGene | 124700 | Electrocompetent cells |
| Genetic reagent (*Burkholderia thailandensis*) | WT with I-*Sce*I cut sites 50 kb outside duplicating region | This study | | Inserted I-*Sce*I recognition sites between LO74_RS08325 and LO74_RS08330 and LO74_RS09660 and LO74_RS09665. |
| Genetic reagent (*Burkholderia thailandensis*) | WT with *gusA* insertion | This study | | *gusA* inserted between LO74_RS09420 and LO74_RS31810 near junction sequence. |
| Genetic reagent (*Burkholderia thailandensis*) | Δ*recA* with *gusA* insertion | This study | | *gusA* inserted between LO74_RS09420 and LO74_RS31810 near junction sequence. Fused first 31 amino acids to last 38 amino acids of LO74_RS12520 (*recA*). |
| Genetic reagent (*Burkholderia thailandensis*) | Δ*recA att*Tn7::*recA* with *gusA* insertion | This study | | *gusA* inserted between LO74_RS09420 and LO74_RS31810 near junction sequence. Fused first 31 amino acids to last 38 amino acids of LO74_RS12520. *recA* inserted at both *att*Tn7 sites. |
| Genetic reagent (*Burkholderia thailandensis*) | $P_{S12}$-*bcpAIOB* Dup⁺ with I-*Sce*I sites 50 kb outside duplicating region | This study | | $P_{S12}$ and nptII inserted between bcpA and its native promoter. *nptII* excised with Flp recombinase. Inserted I-*Sce*I recognition sites between LO74_RS08325 and LO74_RS08330 and LO74_RS09660 and LO74_RS09665. |
| Genetic reagent (*Burkholderia thailandensis*) | Δ*bcpAIOB* with I-*Sce*I sites 50 kb outside duplicating region | This study | | LO74_RS09295–LO74_RS09305 (*bcpAIOB*) replaced with *nptII* surrounded by FRT sites. *nptII* excised with Flp recombinase. Inserted I-*Sce*I recognition sites between LO74_RS08325 and LO74_RS08330 and LO74_RS09660 and LO74_RS09665. |
| Genetic reagent (*Burkholderia thailandensis*) | $P_{S12}$-*bcpAIOB* Dup⁺ | *Anderson et al., 2012* | | $P_{S12}$ promoter inserted between LO74_RS09305 (*bcpA*) and its native promoter. |
| Genetic reagent (*Burkholderia thailandensis*) | Δ*bcpAIOB*::*nptII* Dup⁻ | *Anderson et al., 2012* | | LO74_RS09295–LO74_RS09305 (*bcpAIOB*) replaced with *nptII* surrounded by FRT sites. |
| Genetic reagent (*Burkholderia thailandensis*) | $P_{S12}$-*bcpAIOB* Dup⁻ | This study | | $P_{S12}$ promoter inserted between LO74_RS09305 (*bcpA*) and its native promoter. |
| Genetic reagent (*Burkholderia thailandensis*) | Δ*bcpAIOB*::*ble* Δ*bcpAIOB*::*nptII* Dup⁺ | This study | | LO74_RS09295–LO74_RS09305 (*bcpAIOB*) replaced with *nptII* surrounded by FRT sites. Additional copy of LO74_RS09295– LO74_RS09305 (*bcpAIOB*) replaced with *ble*. |

*Continued on next page*

*Continued*

| Reagent type (species) or resource | Designation | Source or reference | Identifiers | Additional information |
|---|---|---|---|---|
| Genetic reagent (*Burkholderia thailandensis*) | ΔISβ::*nptII* | This study | | Inverted repeats and LO74_RS09425 (ISβ) replaced with *nptII* surrounded by FRT sites. |
| Genetic reagent (*Burkholderia thailandensis*) | ΔISα::*nptII* | This study | | Inverted repeats and LO74_RS28540 (ISα) replaced with *nptII* surrounded by FRT sites. |
| Genetic reagent (*Burkholderia thailandensis*) | ISβ Δ*orfAB*::*nptII* ISα Δ*orfAB* | This study | | LO74_RS28540 (ISα *orfAB*) replaced with *nptII* surrounded by FRT sites. *nptII* excised with Flp recombinase. LO74_RS09425 (ISβ *orfAB*) replaced with *nptII* surrounded by FRT sites. |
| Genetic reagent (*Burkholderia thailandensis*) | Δ*recA* | This study | | Fused first 31 amino acids to last 38 amino acids of LO74_RS12520 (*recA*). |
| Genetic reagent (*Burkholderia thailandensis*) | Δ*recA att*Tn7::*recA* | This study | | Fused first 31 amino acids to last 38 amino acids of LO74_RS1252. *recA* inserted at both *att*Tn7 sites. |
| Genetic reagent (*Burkholderia thailandensis*) | Fragmented *nptII* reporter 500 bp homology | This study | | Inverted repeats and LO74_RS28540 (ISα) replaced with *tet* and bases 46–795 of *nptII* (fwd). Inverted repeats and LO74_RS09425 (ISβ) replaced with *ble* and bases 1–545 of *nptII* (fwd). |
| Genetic reagent (*Burkholderia thailandensis*) | Fragmented *nptII* reporter 150 bp homology | This study | | Inverted repeats and LO74_RS28540 (ISα) replaced with *tet* and bases 46–795 of *nptII* (fwd). Inverted repeats and LO74_RS09425 (ISβ) replaced with *ble* and bases 1–195 of *nptII* (fwd). |
| Genetic reagent (*Burkholderia thailandensis*) | Fragmented *nptII* reporter 50 bp homology | This study | | Inverted repeats and LO74_RS28540 (ISα) replaced with *tet* and bases 46–795 of *nptII* (fwd). Inverted repeats and LO74_RS09425 (ISβ) replaced with *ble* and bases 1–95 of *nptII* (fwd). |
| Genetic reagent (*Burkholderia thailandensis*) | Fragmented *nptII* reporter 0 bp homology | This study | | Inverted repeats and LO74_RS28540 (ISα) replaced with *tet* and bases 46–795 of *nptII* (fwd). Inverted repeats and LO74_RS09425 (ISβ) replaced with *ble* and bases 1–45 of *nptII* (fwd). |
| Genetic reagent (*Burkholderia thailandensis*) | Fragmented *gfp* reporter *att*Tn7::*rfp* | This study | | $P_{S12}$-*rfp* inserted an *att*Tn7 site. Inverted repeats and LO74_RS28540 (ISα) replaced with *tet* and bases 1–589 of *gfp* (rev). Inverted repeats and LO74_RS09425 (ISβ) replaced with *ble* and bases 89–744 of *gfp* (rev). 500 bp of homology in *gfp* fragments. |
| Genetic reagent (*Burkholderia thailandensis*) | Fragmented *gfp* reporter *att*Tn7::*rfp* locked | This study | | $P_{S12}$-*rfp* inserted an *att*Tn7 site. Inverted repeats and LO74_RS28540 (ISα) replaced with *tet* and bases 1–589 of *gfp* (rev). |
| Genetic reagent (*Burkholderia thailandensis*) | Fragmented *gusA* reporter | This study | | Inverted repeats and LO74_RS28540 (ISα) replaced with *tet* and bases 118–1691 of *gusA* (fwd). Inverted repeats and LO74_RS09425 (ISβ) replaced with *ble* and bases 1–1454 of *gusA* (fwd) 1332 bp of homology in *gusA* fragments. |
| Genetic reagent (*Burkholderia thailandensis*) | Fragmented *gusA* reporter locked | This study | | Inverted repeats and LO74_RS28540 (ISα) replaced with *tet* and bases 118–1691 of *gusA* (fwd). |
| Recombinant DNA reagent (plasmid) | pLL31 | This study | | I-SceI recognition sequence and FRT-*nptII*-FRT flanked by 500 bp sequences homologous to portions of LO74_RS08325 and LO74_RS08330. |
| Recombinant DNA reagent (plasmid) | pLL32 | This study | | I-SceI recognition sequence and *ble* flanked by 500 bp sequences homologous to portions of LO74_RS09660 and LO74_RS09665. |

*Continued on next page*

*Continued*

| Reagent type (species) or resource | Designation | Source or reference | Identifiers | Additional information |
|---|---|---|---|---|
| Recombinant DNA reagent (plasmid) | pLL44 | This study | | *gusA* and *ble* flanked by 500 bp homologous to sequences within LO74_RS09420 and between LO74_RS09420 and LO74_RS31810. |
| Recombinant DNA reagent (plasmid) | pABT84 | This study | | FRT-*nptII*-FRT flanked by 500 bp sequences homologous to LO74_RS12515 and LO74_RS12525. |
| Recombinant DNA reagent (plasmid) | pLL46 | This study | | *recA* under control of its native promoter |
| Recombinant DNA reagent (plasmid) | pTNS3 | *Choi et al., 2008* | | Helper plasmid for mini-Tn7, oriT, R6K ori |
| Recombinant DNA reagent (plasmid) | pABT86 | This study | | $P_{S12}$ promoter and FRT-*nptII*-FRT flanked by 500 bp sequences homologous to LO74_RS09305 and LO74_RS28675. |
| Recombinant DNA reagent (plasmid) | pFlpTet | *Garcia et al., 2013* | | Rham-inducible *flp*, TS ori |
| Recombinant DNA reagent (plasmid) | pLL37 | This study | | ble flanked by 500 bp sequences homologous to LO74_RS09295- LO74_RS09305. |
| Recombinant DNA reagent (plasmid) | pLL38 | This study | | FRT-*nptII*-FRT flanked by 500 bp sequences homologous to LO74_RS09295- LO74_RS09305. |
| Recombinant DNA reagent (plasmid) | pLL29 | This study | | FRT-*nptII*-FRT flanked by 500 bp sequences homologous to sequences between LO74_RS31810 and LO74_RS09435 and LO74_RS09435. |
| Recombinant DNA reagent (plasmid) | pABT104 | This study | | FRT-*nptII*-FRT flanked by 500 bp sequences homologous to LO74_RS08585 and LO74_RS08600. |
| Recombinant DNA reagent (plasmid) | pABT66 | *Ocasio and Cotter, 2019* | | FRT-*nptII*-FRT flanked by 500 bp sequences homologous to LO74_RS31810 through 5' ISβ inverted repeat and 3' ISβ inverted repeat through LO74_RS09435. |
| Recombinant DNA reagent (plasmid) | pABT78 | *Ocasio and Cotter, 2019* | | FRT-*nptII*-FRT flanked by 500 bp sequences homologous to LO74_RS08585 through 5' ISα inverted repeat and 3' ISα inverted repeat through LO74_RS08600. |
| Recombinant DNA reagent (plasmid) | pLL61 | This study | | *tet* and *nptII*$_{46–795}$ (fwd) flanked by 500 bp sequences homologous to LO74_RS08585 and LO74_RS08600. |
| Recombinant DNA reagent (plasmid) | pLL62 | This study | | *ble* and *nptII*$_{1–545}$ (fwd) flanked by 500 bp sequences homologous to LO74_RS09420 and LO74_RS31810 and LO74_RS09435. |
| Recombinant DNA reagent (plasmid) | pLL63 | This study | | *ble* and *nptII*$_{1–195}$ (fwd) flanked by 500 bp sequences homologous to LO74_RS09420 and LO74_RS31810 and LO74_RS09435. |
| Recombinant DNA reagent (plasmid) | pLL64 | This study | | *ble* and *nptII*$_{1–95}$ (fwd) flanked by 500 bp sequences homologous to LO74_RS09420 and LO74_RS31810 and LO74_RS09435. |
| Recombinant DNA reagent (plasmid) | pLL74 | This study | | *ble* and *nptII*$_{1–45}$ (fwd) flanked by 500 bp sequences homologous to LO74_RS09420 and LO74_RS31810 and LO74_RS09435. |
| Recombinant DNA reagent (plasmid) | pLL72 | This study | | *tet* and *gusA*$_{118–1691}$ (fwd) flanked by 500 bp sequences homologous to LO74_RS08585 and LO74_RS08600. |
| Recombinant DNA reagent (plasmid) | pLL73 | This study | | *tet* and *gusA*$_{1–1454}$ (fwd) flanked by 500 bp sequences homologous to LO74_RS09420 and LO74_RS31810 and LO74_RS09435. |

*Continued on next page*

*Continued*

| Reagent type (species) or resource | Designation | Source or reference | Identifiers | Additional information |
|---|---|---|---|---|
| Recombinant DNA reagent (plasmid) | mini-Tn7-km-rfp | *Norris et al., 2010* | | P$_{S12}$-Turbo*rfp* |
| Recombinant DNA reagent (plasmid) | pLL65 | This study | | *tet* and *gfp*$_{1–589}$ (rev) flanked by 500 bp sequences homologous to LO74_RS08585 and LO74_RS08600. |
| Recombinant DNA reagent (plasmid) | pLL66 | This study | | *tet* and *gusA*$_{89–744}$ (rev) flanked by 500 bp sequences homologous to LO74_RS09420 and LO74_RS31810 and LO74_RS09435. |
| Sequence-based reagent | Junc1 | IDT | | 5' GCCGTGCTAGAGAGGCGCTA 3' |
| Sequence-based reagent | Junc2 | IDT | | 5' AGCAGAATCAGATGCACGCCATTCG 3' |
| Sequence-based reagent | Ctrl1 | IDT | | 5' TGATGCAGTTTCCGGCGCAGTAAC 3' |
| Sequence-based reagent | Ctrl2 | IDT | | 5' AATCGTGTCGGCGTGTGACGAA 3' |
| Chemical compound | X-Gluc (5-bromo-4-chloro-3-indolyl-beta-D-glucuronic acid, cyclohexyl ammonium salt) | Goldbio | B-735-250 | |
| Chemical compound, drug | Kanamycin Monosulfate | Chem-Impex International, Inc | 00195 | |
| Software | GraphPad Prism | GraphPad Prism (https://graphpad.com) | | Version 9.5.0 |
| Software | Imaris x64 | Imaris (https://imaris.oxinst.com/) | | Version 9.9.1 |

## Bacterial strains and culture conditions

*Bt*E264 is an environmental isolate (*Brett et al., 1998*). Strains 2002721643 and 2002721723 were obtained from the Department of Defense's Unified Culture Collection (*Johnson et al., 2015*). Plasmids were maintained in *E. coli* DH5α. For insertion at the *att*Tn7 sites or Flp-mediated FRT recombination, plasmids were introduced into *Bt*E264 by conjugation with *E. coli* donor strain, RHO3 (*López et al., 2009*). *Bt*E264 and *E. coli* strains were grown overnight with aeration at 37°C (unless indicated) in low-salt Luria-Bertani (LSLB, 0.5% NaCl). Antibiotics and supplements were added to cultures at the following concentrations: 50 µg/ml X-Gluc (5-bromo-4-chloro-3-indoxyl-beta-D-glucuronide), 200 µg/ml 2,6-diaminopimelic acid (DAP), 0.2% (wt/vol) rhamnose, 500 µg/ml (for *Bt*E264) or 50 µg/ml (for *E. coli*) kanamycin (Km), 100 µg/ml ampicillin, or 200 µg/ml zeocin as appropriate. When indicated, BtE264 was cultured in M63 minimal medium (110 mM KH$_2$PO$_4$, 200 mM K$_2$HPO$_4$, 75 mM (NH$_4$)$_2$SO$_4$, 16 nM FeSO$_4$) supplemented with 1 mM MgSO$_4$ and 0.2% glucose. For passaging experiments, M63 was further supplemented with 0.4% glycerol and 0.01% casamino acids.

## Mutant construction techniques

To introduce a mutation by natural transformation, we linearized plasmids containing a gene-encoding antibiotic resistance and additional sequences, if necessary, flanked by ~500 bp sequences with homology to genomic regions of interest. Natural transformation was conducted following previously described protocols (*Thongdee et al., 2008*). Transformants were selected on LSLB containing the appropriate antibiotic and verified through PCR analysis.

Insertion of sequences to one or both *att*Tn7 sites was conducted with triparental mating between *Bt*E264, *E. coli* RHO3 containing the helper plasmid pTNS3, and *E. coli* RHO3 carrying a plasmid with sequences of interest according to previously described protocols (*Choi et al., 2005*). Mutants were confirmed with PCR analysis.

Removal of antibiotic resistance cassettes flanked by FRT sites was carried out through Flp-mediated FRT recombination according to previously described protocols (*Choi et al., 2008*).

## Plasmids

| Name | Information | Resistance marker | Backbone | First appeared |
|------|-------------|-------------------|----------|----------------|
| pLL31 | I-SceI recognition sequence and FRT-*nptII*-FRT flanked by 500 bp sequences homologous to portions of LO74_RS08325 and LO74_RS08330 | Kanamycin, Ampicillin | pTWIST Amp High copy | This study |
| pLL32 | I-SceI recognition sequence and *ble* flanked by 500 bp sequences homologous to portions of LO74_RS09660 and LO74_RS09665 | Zeocin, Ampicillin | pJET2.1 | This study |
| pLL44 | *gusA* and *ble* flanked by 500 bp homologous to sequences within LO74_RS09420 and between LO74_RS09420 and LO74_RS31810 | Zeocin, Ampicillin | pJET2.1 | This study |
| pABT84 | FRT-*nptII*-FRT flanked by 500 bp sequences homologous to LO74_RS12515 and LO74_RS12525 | Kanamycin, Ampicillin | pJET2.1 | This study |
| pLL46 | *recA* under control of its native promoter | Tetracycline | pUC18tet | This study |
| pTNS3 | Helper plasmid for mini-Tn7, oriT, R6K ori | Ampicillin | | *Choi et al., 2008* |
| pABT86 | P$_{S12}$ promoter and FRT-*nptII*-FRT flanked by 500 bp sequences homologous to LO74_RS09305 and LO74_RS28675 | Kanamycin, Ampicillin | pJET2.1 | This study |
| pFlpTet | Rham-inducible *flp*, TS ori | Tetracycline | | *Garcia et al., 2013* |
| pLL37 | ble flanked by 500 bp sequences homologous to LO74_RS09295-LO74_RS09305 | Zeocin, Ampicillin | pJET2.1 | This study |
| pLL38 | FRT-*nptII*-FRT flanked by 500 bp sequences homologous to LO74_RS09295–LO74_RS09305 | Kanamycin, Ampicillin | pJET2.1 | This study |
| pLL29 | FRT-*nptII*-FRT flanked by 500 bp sequences homologous to sequences between LO74_RS31810 and LO74_RS09435 and LO74_RS09435 | Kanamycin, Ampicillin | pJET2.1 | This study |

*Continued on next page*

*Continued*

| Name | Information | Resistance marker | Backbone | First appeared |
|---|---|---|---|---|
| pABT104 | FRT-*nptII*-FRT flanked by 500 bp sequences homologous to LO74_RS08585 and LO74_RS08600 | Kanamycin, Ampicillin | pJET2.1 | This study |
| pABT66 | FRT-*nptII*-FRT flanked by 500 bp sequences homologous to LO74_RS31810 through 5′ISβ inverted repeat and 3′ISβ inverted repeat through LO74_RS09435 | Kanamycin, Ampicillin | pJET2.1 | *Ocasio and Cotter, 2019* |
| pABT78 | FRT-*nptII*-FRT flanked by 500 bp sequences homologous to LO74_RS08585 through 5′ISα inverted repeat and 3′ISα inverted repeat through LO74_RS08600 | Kanamycin, Ampicillin | pJET2.1 | *Ocasio and Cotter, 2019* |
| pLL61 | *tet* and $nptII_{46-795}$ (fwd) flanked by 500 bp sequences homologous to LO74_RS08585 and LO74_RS08600 | Tetracycline, Ampicillin | pTWIST Amp High copy | This study |
| pLL62 | *ble* and $nptII_{1-545}$ (fwd) flanked by 500 bp sequences homologous to LO74_RS09420 and LO74_RS31810 and LO74_RS09435 | Zeocin, Ampicillin | pJET2.1 | This study |
| pLL63 | *ble* and $nptII_{1-195}$ (fwd) flanked by 500 bp sequences homologous to LO74_RS09420 and LO74_RS31810 and LO74_RS09435 | Zeocin, Ampicillin | pJET2.1 | This study |
| pLL64 | *ble* and $nptII_{1-95}$ (fwd) flanked by 500 bp sequences homologous to LO74_RS09420 and LO74_RS31810 and LO74_RS09435 | Zeocin, Ampicillin | pJET2.1 | This study |
| pLL74 | *ble* and $nptII_{1-45}$ (fwd) flanked by 500 bp sequences homologous to LO74_RS09420 and LO74_RS31810 and LO74_RS09435 | Zeocin, Ampicillin | pJET2.1 | This study |
| pLL72 | *tet* and $gusA_{118-1691}$ (fwd) flanked by 500 bp sequences homologous to LO74_RS08585 and LO74_RS08600 | Tetracycline, Ampicillin | pTWIST Amp High copy | This study |
| pLL73 | *tet* and $gusA_{1-1454}$ (fwd) flanked by 500 bp sequences homologous to LO74_RS09420 and LO74_RS31810 and LO74_RS09435 | Zeocin, Ampicillin | pJET2.1 | This study |

*Continued*

| Name | Information | Resistance marker | Backbone | First appeared |
|------|-------------|-------------------|----------|----------------|
| mini-Tn7-km-rfp | P$_{S12}$-Turbo*rfp* | Kanamycin, Ampicillin | pUC18km | ***Norris et al., 2010*** |
| pLL65 | *tet* and *gfp*$_{1-589}$ (rev) flanked by 500 bp sequences homologous to LO74_RS08585 and LO74_RS08600 | Tetracycline, Ampicillin | pTWIST Amp High copy | This study |
| pLL66 | *tet* and *gusA*$_{89-744}$ (rev) flanked by 500 bp sequences homologous to LO74_RS09420 and LO74_RS31810 and LO74_RS09435 | Zeocin, Ampicillin | pJET2.1 | This study |

## Mutants and mutant construction

| Name | Information | Plasmid(s) used | Construction method | Resistance(s) | Figure referenced |
|------|-------------|-----------------|---------------------|---------------|-------------------|
| WT with I-*Sce*I cut sites 50 kb outside duplicating region | Inserted I-*Sce*I recognition sites between LO74_RS08325 and LO74_RS08330 and LO74_RS09660 and LO74_RS09665. | pLL31, pLL32 | Two rounds of natural transformation | Kanamycin, Zeocin | 3A, 3B, 6A |
| WT with *gusA* insertion | *gusA* inserted between LO74_RS09420 and LO74_RS31810 near junction sequence. | pLL44 | Natural transformation | Zeocin | 4C, 4D |
| Δ*recA* with *gusA* insertion | *gusA* inserted between LO74_RS09420 and LO74_RS31810 near junction sequence. Fused first 31 amino acids to last 38 amino acids of LO74_RS12520 (*recA*). | pLL44, pABT84 | Two rounds of natural transformation | Zeocin, Kanamycin | 4D |
| Δ*recA att*Tn7::*recA* with *gusA* insertion | *gusA* inserted between LO74_RS09420 and LO74_RS31810 near junction sequence. Fused first 31 amino acids to last 38 amino acids of LO74_RS12520. *recA* inserted at both *att*Tn7 sites. | pLL44, pABT84, pTNS3, pLL46 | Two rounds of natural transformation and triparental mating | Zeocin, Kanamycin, Tetracycline | 4D |

*Continued on next page*

*Continued*

| Name | Information | Plasmid(s) used | Construction method | Resistance(s) | Figure referenced |
|---|---|---|---|---|---|
| P$_{S12}$-*bcpAIOB* Dup$^+$ with I-*Sce*I sites 50 kb outside duplicating region | P$_{S12}$ and nptII inserted between *bcpA* and its native promoter. *nptII* excised with Flp recombinase. Inserted I-*Sce*I recognition sites between LO74_RS08325 and LO74_RS08330 and LO74_RS09660 and LO74_RS09665. | pABT86, pFlpTet, pLL31, pLL32 | Natural transformation, Flp-FRT recombination, two rounds of natural transformation | Kanamycin, Zeocin | 6A |
| Δ*bcpAIOB* with I-*Sce*I sites 50 kb outside duplicating region | LO74_RS09295-LO74_RS09305 (*bcpAIOB*) replaced with *nptII* surrounded by FRT sites. *nptII* excised with Flp recombinase. Inserted I-*Sce*I recognition sites between LO74_RS08325 and LO74_RS08330 and LO74_RS09660 and LO74_RS09665. | pLL38, pFlpTet, pLL31, pLL32 | Natural transformation, Flp-FRT recombination, two rounds of natural transformation | Kanamycin, Zeocin | 6A |
| P$_{S12}$-*bcpAIOB* Dup$^-$ | P$_{S12}$ promoter inserted between LO74_RS09305 (*bcpA*) and its native promoter. | pABT86 | Natural transformation | Kanamycin | 6C |
| Δ*bcpAIOB*::*ble* Δ*bcpAIOB*::*nptII* Dup$^+$ | LO74_RS09295–LO74_RS09305 (*bcpAIOB*) replaced with *nptII* surrounded by FRT sites. Additional copy of LO74_RS09295–LO74_RS09305 (*bcpAIOB*) replaced with *ble*. | pLL37, pLL38 | Two rounds of natural transformation | Zeocin, Kanamycin | 6D |
| ΔISβ::*nptII* | Inverted repeats and LO74_RS09425 (ISβ) replaced with *nptII* surrounded by FRT sites. | pLL29 | Natural transformation | Kanamycin | 6E |
| ΔISα::*nptII* | Inverted repeats and LO74_RS28540 (ISα) replaced with *nptII* surrounded by FRT sites. | pABT104 | Natural transformation | Kanamycin | 6E |

*Continued*

| Name | Information | Plasmid(s) used | Construction method | Resistance(s) | Figure referenced |
|---|---|---|---|---|---|
| ISβ Δ*orfAB*::*nptII* ISα Δ*orfAB* | LO74_RS28540 (ISα *orfAB*) replaced with *nptII* surrounded by FRT sites. *nptII* excised with Flp recombinase. LO74_RS09425 (ISβ *orfAB*) replaced with *nptII* surrounded by FRT sites. | pABT66, pFlptet, pABT78 | Natural transformation, Flp-FRT recombination, natural transformation | Kanamycin | 6F |
| Δ*recA* | Fused first 31 amino acids to last 38 amino acids of LO74_RS12520 (*recA*). | pABT84 | Natural transformation | Kanamycin | 7A |
| Δ*recA att*Tn7::*recA* | Fused first 31 amino acids to last 38 amino acids of LO74_RS1252. *recA* inserted at both *att*Tn7 sites. | pABT84, pTNS3, pLL46 | Natural transformation and triparental mating | Kanamycin, Tetracycline | 7A |
| Fragmented *nptII* reporter 500 bp homology | Inverted repeats and LO74_RS28540 (ISα) replaced with *tet* and bases 46–795 of *nptII* (fwd). Inverted repeats and LO74_RS09425 (ISβ) replaced with *ble* and bases 1–545 of *nptII* (fwd). | pLL61, pLL62 | Two rounds of natural transformation | Tetracycline, Zeocin | 7B, 7D, 7E |
| Fragmented *nptII* reporter 150 bp homology | Inverted repeats and LO74_RS28540 (ISα) replaced with *tet* and bases 46–795 of *nptII* (fwd). Inverted repeats and LO74_RS09425 (ISβ) replaced with *ble* and bases 1–195 of *nptII* (fwd). | pLL61, pLL63 | Two rounds of natural transformation | Tetracycline, Zeocin | 7B, 7D |
| Fragmented *nptII* reporter 50 bp homology | Inverted repeats and LO74_RS28540 (ISα) replaced with *tet* and bases 46–795 of *nptII* (fwd). Inverted repeats and LO74_RS09425 (ISβ) replaced with *ble* and bases 1–95 of *nptII* (fwd). | pLL61, pLL64 | Two rounds of natural transformation | Tetracycline, Zeocin | 7B, 7D |

*Continued*

| Name | Information | Plasmid(s) used | Construction method | Resistance(s) | Figure referenced |
|---|---|---|---|---|---|
| Fragmented *nptII* reporter 0 bp homology | Inverted repeats and LO74_RS28540 (ISα) replaced with *tet* and bases 46–795 of *nptII* (fwd). Inverted repeats and LO74_RS09425 (ISβ) replaced with *ble* and bases 1–45 of *nptII* (fwd). | pLL61, pLL74 | Two rounds of natural transformation | Tetracycline, Zeocin | 7D |
| Fragmented *gfp* reporter *att*Tn7::*rfp* | $P_{S12}$-*rfp* inserted an *att*Tn7 site. Inverted repeats and LO74_RS28540 (ISα) replaced with *tet* and bases 1–589 of *gfp* (rev). Inverted repeats and LO74_RS09425 (ISβ) replaced with *ble* and bases 89–744 of *gfp* (rev). 500 bp of homology in *gfp* fragments. | pLL65, pLL66, pTNS3, mini-Tn7-km-rfp | Triparental mating and two rounds of natural transformation | Tetracycline, Zeocin | 8A, 8B, 8C, 8D |
| Fragmented *gfp* reporter *att*Tn7::*rfp* locked | $P_{S12}$-*rfp* inserted an *att*Tn7 site. Inverted repeats and LO74_RS28540 (ISα) replaced with *tet* and bases 1–589 of *gfp* (rev). | pLL65, pTNS3, mini-Tn7-km-rfp | Triparental mating and natural transformation | Tetracycline | 8B, 8C, 8D |
| Fragmented *gusA* reporter | Inverted repeats and LO74_RS28540 (ISα) replaced with *tet* and bases 118–1691 of *gusA* (fwd). Inverted repeats and LO74_RS09425 (ISβ) replaced with *ble* and bases 1–1454 of *gusA* (fwd) 1332 bp of homology in *gusA* fragments. | pLL72, pLL73 | Two rounds of natural transformation | Tetracycline, Zeocin | 9A, 9B, 9C, 9D, 9E |
| Fragmented *gusA* reporter locked | Inverted repeats and LO74_RS28540 (ISα) replaced with *tet* and bases 118–1691 of *gusA* (fwd). | pLL72 | Two rounds of natural transformation | Tetracycline | 9C, 9D |

## Polymerase chain reaction (PCR) analysis

Junction PCR analyses were conducted as described previously (*Ocasio and Cotter, 2019*) using Junc1 (5′ GCCGTGCTAGAGAGGCGCTA 3′) and Junc2 (5′ AGCAGAATCAGATGCACGCCATTCG 3′) to amplify the junction sequence and Ctrl1 (5′ TGATGCAGTTTCCGGCGCAGTAAC 3′) and Ctrl2 (5′ AATCGTGTCGGCGTGTGACGAA 3′) to amplify a fragment inside the region (BTH_I2728) as a positive control. When necessary, PCR reactions were visualized by gel electrophoresis within 0.8% gels alongside 1 kb HyperLadder as a size standard.

## Duplication-dependent phenotypic analysis

Colony morphologies were analyzed after 48 hr growth on LSLB agar and imaged at ×0.63 magnification on an Olympus MVX10 macroscope.

Test tube aggregation was assayed as previously described (*Ocasio and Cotter, 2019*).

Colony biofilms shown in *Figures 2D and 6C, D* were created by spotting 20 μl of an overnight culture diluted to an $OD_{600}$ of 0.2 on LSLB agar. Biofilms were incubated at room temperature for 28 days and imaged at ×0.63 magnification on an Olympus MVX10 macroscope.

Colony biofilms shown in *Figure 5A* were created by spotting 10 μl of an overnight culture diluted to an $OD_{600}$ of 0.2 on LSLB agar. Biofilms were incubated at room temperature for 30 days and imaged at ×0.63 magnification on an Olympus MVX10 macroscope.

For the results shown in *Figures 5B, C and 6B, E, F*, colony biofilms were created by spotting 20 μl of an overnight culture diluted to an $OD_{600}$ of 1.0 on LSLB agar. Biofilms were incubated at room temperature for 7 days. Duplication formation in the center and edge of the colony biofilm (*Figure 4B*) was analyzed by PCR.

For the experiments, results in *Figure 7E*, colony biofilms were created by spotting 20 μl of an overnight culture diluted to an $OD_{600}$ of 1.0 on LSLB agar. Biofilms were incubated at room temperature for up to 22 days. Every other day, bacteria were picked from the center and edge of the colony biofilm (*Figure 4B*), serially diluted to $10^9$ and plated on LSLB agar and LSLB agar containing Km. After 16–18 hr of incubation at 37°C, colonies were counted and cfu was calculated on both media. The proportion on $Km^R$ colonies was calculated by dividing the cfu of $Km^R$ colonies by the total cfu.

## Pulsed-field gel electrophoresis

Cell pellets from 2 ml overnight cultures were used to create gel plugs using the CHEF Bacterial Genomic DNA Plug Kit (Bio-Rad). DNA was digested with I-SceI. DNA fragments were resolved with 1% certified megabase agarose (Bio-Rad) containing GelRed Nucleic Acid Gel Stain (Biotium) in 0.5× Tris-borate_EDTA buffer (TBE). For *Figure 3B*, gels were run for 24 hr at 14°C, 6 V/cm, included angel of 120° with switch times gradually ascending from 75 to 155 s. For *Figure 6A*, gels were run for 24 hr at 14°C, 6 V/cm, included angel of 120° with switch times gradually ascending from 70 to 150 s. For *Figure 3—figure supplement 1B*, gels were run for 48 hr at 14°C, 3.6 V/cm, included angel of 120° with switch times gradually ascending from 50 to 60 s. Lambda ladder was used as a size standard.

## Passage experiments

For the experiment, results in *Figure 4D*, overnight cultures of the *gusA* insertional strain in LSLB were washed in phosphate-buffered saline (PBS) and diluted down to an $OD_{600}$ of 0.02 in 3 ml of LSLB. Following 24 hr of growth with aeration, an aliquot was subcultured at a $OD_{600}$ of 0.02 in LSLB. Passaging was repeated every 24 hr until the end of the experiment. Every 5 days, an aliquot was diluted and plated for individual colonies on LSLB containing X-Gluc. The number of blue and white colonies was counted after the plate was incubated at 37°C for 24 hr and room temp for 48 hr.

For the experiment, results in *Figure 7D*, overnight cultures in LSLB of the fragmented *nptII* reporter strain were serially diluted to $10^9$. Dilutions were plated on both LSLB agar and LSLB agar containing Km. After 16–18 hr of incubation at 37°C, colonies were counted and cfu was calculated for both media. The proportion on $Km^R$ colonies was calculated by dividing the cfu of $Km^R$ colonies by the total cfu.

For the experiment, results in *Figure 9B–E*, overnight cultures of the $Dup^+$, $Dup^-$, and $Dup^-$-locked fragmented *gusA* reporter strains in LSLB were washed in PBS and diluted down to an $OD_{600}$ of 0.2 in 3 ml of M63. Following 24 hr of growth with aeration, an aliquot was subculture at a $OD_{600}$ of 0.2 in 3 ml of M63 for the liquid passage. For the biofilm passage, spent media was removed and 3 ml of

fresh M63 was returned to the same tube. Passaging was repeated every 24 hr until the end of the experiment. Every 5 days, an aliquot was diluted and plated for individual colonies on LSLB containing X-Gluc. The number of blue and white colonies was counted after the plate was incubated at 37°C for 24 hr and room temperature for 48 hr.

### Biofilm microscopy

Overnight cultures of fragmented *gfp* reporter strains were diluted to an OD600 of 0.02 in 400 µl of M63 in 4-well chambered coverglass dishes. Biofilms were incubated at 45°C for 24–72 hr in humified chambers. Before imaging, biofilms were washed four times in PBS and overlaid with 400 µl PBS. Biofilms were imaged using confocal laser microscopy using a Zeiss LSM 780 with a ×40 objective. Z-stacks were processed and analyzed using Imaris x64 9.9.1 and the Biofilm Analysis Xtension (Bitplane Scientific Software). Red fluorescence was pseudo colored as magenta.

## Acknowledgements

Thank you to Colin Manoil, Josephine Chandler, Joseph Mougous, and the U.S. Army Medical Research Institute of Infectious Diseases (USAMRIID) for sending us *B. thailandensis* strains. We are also grateful to the members of the Cotter laboratory for their continued support and insightful discussions.

Microscopy was performed at the UNC Neuroscience Microscopy Core (RRID:SCR_019060), supported, in part, by funding from the NIH-NINDS Neuroscience Center Support Grant P30 NS045892 and the NIH-NICHD Intellectual and Developmental Disabilities Research Center Support Grant P50 HD103573. This research was supported by grants from the NIH: R35 GM136533 and R01 GM121110 to P.A.C.

## Additional information

### Funding

| Funder | Grant reference number | Author |
| --- | --- | --- |
| National Institutes of Health | R35 GM136533 | Peggy A Cotter |
| National Institutes of Health | R01 GM121110 | Peggy A Cotter |

The funders had no role in study design, data collection, and interpretation, or the decision to submit the work for publication.

### Author contributions

Lillian C Lowrey, Conceptualization, Data curation, Formal analysis, Investigation, Visualization, Methodology, Writing - original draft, Writing - review and editing; Leslie A Kent, Angelica B Ocasio, Conceptualization, Investigation, Methodology; Bridgett M Rios, Investigation; Peggy A Cotter, Conceptualization, Formal analysis, Funding acquisition, Project administration, Writing - review and editing

### Author ORCIDs

Lillian C Lowrey http://orcid.org/0000-0001-7602-033X
Leslie A Kent http://orcid.org/0000-0003-3195-0247
Peggy A Cotter http://orcid.org/0000-0003-3726-3990

### Decision letter and Author response

Decision letter https://doi.org/10.7554/eLife.84327.sa1
Author response https://doi.org/10.7554/eLife.84327.sa2

## Additional files

### Supplementary files
• MDAR checklist

### Data availability
All data generated or analyzed during this study are included in the manuscript and supporting file; Source Data files have been provided for all gels.

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
