## [Editor Report]

This paper reports a bet hedging strategy in bacteria based on chromosomal duplications and rearrangements that confer advantages in certain growth conditions. The work is of fundamental importance for understanding the role of genetic and biological variation in bacteria. The experimental work is exceptionally strong and convincing. The paper will be of interest to a broad audience including bacteriologists, geneticists and evolutionary biologists.

---

## [Decision Letter]

**Decision letter after peer review:**

Thank you for submitting your article "An IS-mediated, RecA-dependent, bet-hedging strategy in Burkholderia thailandensis" for consideration by *eLife*. Your article has been reviewed by 3 peer reviewers, and the evaluation has been overseen by a Reviewing Editor and Arturo Casadevall as the Senior Editor. The following individual involved in review of your submission have agreed to reveal their identity: Michael T Laub (Reviewer #3).

Essential revisions:

As you can see below all three reviewers were unanimous that this is an outstanding paper. It is nice to see such consensus among reviewers. I hope you enjoy and appreciate the thoughtful comments from all three reviewers. Congratulations on great work! We look forward to seeing the revised manuscript and I anticipate that it will be accepted.

*Reviewer #1 (Recommendations for the authors):*

Data in Figure 6B are alluded to (lines 354-5) but are not described in detail.

*Reviewer #2 (Recommendations for the authors):*

I commend the authors for a beautiful and thorough study. The manuscript is already robust and the authors have done so much to study this intriguing system. I am therefore limiting my thoughts about additional investigation to just two points.

1) A major question that remains is whether the different types of cells have differential impacts during in vivo infection. Which cell type predominates during in vivo infection? For example, if mice were infected with cells from flat, rough, or smooth colonies, are there changes in the ratio of the different cell types after in vivo infection? These results could speak to the functional relevance of this phenomenon beyond in vitro biofilm formation.

2) Flat cells did not transition into rough or smooth cells in the restreaking experiment (Figure 2B). It is likely that flat colonies can yield the other colony types after restreaking but at a lower rate. It would further strengthen the manuscript if the authors can define the rate at which flat cells do transition to rough or smooth cells, or to prove that this does not happen.

*Reviewer #3 (Recommendations for the authors):*

I found very little to criticize and think the paper can be published almost exactly as it reads right now.

---

## [Author Response]

Reviewer #1 (Recommendations for the authors):Data in Figure 6B are alluded to (lines 354-5) but are not described in detail.

Corrected. Text referring to the data in 6B incorrectly referred to 6C. Hopefully, it is now clear that lines 339 – 344 describe Figure 6B.

Reviewer #2 (Recommendations for the authors):I commend the authors for a beautiful and thorough study. The manuscript is already robust and the authors have done so much to study this intriguing system. I am therefore limiting my thoughts about additional investigation to just two points.1) A major question that remains is whether the different types of cells have differential impacts during in vivo infection. Which cell type predominates during in vivo infection? For example, if mice were infected with cells from flat, rough, or smooth colonies, are there changes in the ratio of the different cell types after in vivo infection? These results could speak to the functional relevance of this phenomenon beyond in vitro biofilm formation.

We are certainly interested in understanding where biofilm growth is an advantage for *Burkholderia thailandensis* in nature. However, to our knowledge, there is no evidence that *B. thailandensis* naturally infects animals. Strain BPM that we refer to in the discussion is an unusually pathogenic strain with some similarity to *B. pseudomallei*. In the study describing BPM (Reference 28), BPM and E264 were compared in a mouse infection and E264 was shown to be avirulent. In the interest of limiting the use of animals, therefore, we prefer to not conduct infection experiments with E264.

2) Flat cells did not transition into rough or smooth cells in the restreaking experiment (Figure 2B). It is likely that flat colonies can yield the other colony types after restreaking but at a lower rate. It would further strengthen the manuscript if the authors can define the rate at which flat cells do transition to rough or smooth cells, or to prove that this does not happen.

While we did not observe flat colonies (Dup-) producing rough and smooth daughter colonies (Dup+), we were able to see duplications form when we used techniques that allowed us to select rather than screen for cells with a duplication. As shown in Figure 7D, the rate of duplication for a strain containing 500 bp of homology was between 10^-4^ and 10^-5^ in an overnight culture. Since the IS elements provide ~1.3 kb of homology, we expect the rate would be slightly higher. However, we would likely need to screen at least 10,000 colonies to observe a rough or smooth daughter colony.